# Rab GTPases and phosphoinositides fine-tune SNAREs dependent targeting specificity of intracellular vesicle traffic

Seiichi Koike[1,2] & Reinhard Jahn [1]✉

In the secretory pathway the destination of trafficking vesicles is determined by specific proteins that, with the notable exception of SNAREs, are recruited from soluble pools. Previously we have shown that microinjected proteoliposomes containing early or late endosomal SNAREs, respectively, are targeted to the corresponding endogenous compartments, with targeting specificity being dependent on the recruitment of tethering factors by some of the SNAREs. Here, we show that targeting of SNARE-containing liposomes is refined upon inclusion of polyphosphoinositides and Rab5. Intriguingly, targeting specificity is dependent on the concentration of PtdIns(3)P, and on the recruitment of PtdIns(3)P binding proteins such as rabenosyn-5 and PIKfyve, with conversion of PtdIns(3)P into PtdIns(3,5)P2 re-routing the liposomes towards late endosomes despite the presence of GTP-Rab5 and early endosomal SNAREs. Our data reveal a complex interplay between permissive and inhibitory targeting signals that sharpen a basic targeting and fusion machinery for conveying selectivity in intracellular membrane traffic.

Eukaryotic cells contain membranous organelles that dynamically exchange trafficking vesicles, which is essential for cell homeostasis and functioning[1]. Modern imaging approaches revealed that vesicle traffic is of stunning complexity, with thousands of transport vesicles being *en-route* at any given time in a typical mammalian cell[2–4]. To reach the correct target compartment, trafficking vesicles are equipped with unique identifiers (molecular "zip-codes") that form combinatorial codes and recruit sets of effectors, which together mediate recognition, docking, and finally fusion with the target compartment. Molecular zip codes are imprinted during budding where the vesicle composition is controlled by sorting, frequently mediated by specific coats and adapters[5]. When vesicles are isolated from cells and fusion is monitored in vitro, targeting specificity is usually maintained[6,7]. Thus, molecular zip codes are stably associated with the membrane and suffice to convey targeting specificity.

It is widely accepted that there are two major classes of zip code molecules: small GTPases and phosphorylated variants of the membrane lipid phosphatidylinositol (PtdInsP_x). The GTPases involved in trafficking mostly belong to the Rab/Ypt and Arf/Arl subclasses[8,9], each of which representing large protein families (in humans 63 Rabs and 29 Arf/Arls have been identified). These GTPases share common switching mechanisms but differ in the spectra of effector proteins recruited upon activation. Thus, the GTPases provide sufficient diversity for unique tagging of each trafficking pathway, and although some Rabs may operate at multiple trafficking steps[10], they are presently being considered as the main determinants for specificity[11]. The GTPases are switched on by nucleotide exchange factors (GEFs) and switched off by GTPase activating proteins (GAPs)[9]. Moreover, membrane binding of the GTPases, mediated by prenyl moieties or switchable amphiphilic helices, is reversible, with membrane association and dissociation being usually linked to GTP-GDP cycle[8,9]. Furthermore, during ongoing membrane traffic Rabs may work consecutively in a "hand-over-hand" mechanism, with a Rab responsible for a given trafficking step recruiting the GEF for the Rab required for the subsequent trafficking step, thus guaranteeing ordered and

[1]Laboratory of Neurobiology, Max Planck Institute for Multidisciplinary Sciences, Göttingen, Germany. [2]Present address: University of Toyama, Laboratory of Molecular and Cellular Biology, Department of Life Sciences and Bioengineering, 3190 Gofuku, Toyama City 930-8555, Japan. ✉e-mail: rjahn@mpinat.mpg.de

vectorial progression through the pathway[12,13]. Mis-localization or other interference with Rab function results in perturbation of vesicle traffic[14–17].

Phosphoinositides (PtdInsP$_x$) have emerged as being essential in defining membrane identity and thus constitute the second class of zip code molecules[18,19]. The inositol ring can be phosphorylated at any of three hydroxyl groups, giving rise to a total of 7 different PtdInsP$_x$-species that are each associated with distinct intracellular compartments[18]. Similar to the GTPases, the PtdInsP$_x$ variants recruit sets of effector proteins via specific PtdInsP$_x$ binding domains, frequently in cooperation with other signals such as GTPases[18]. Switching is regulated by site-specific PtdIns-kinases ("on"-switch) and PtdIns-phosphatases ("off"-switch), but conversion from one to another active species is also easily possible. Genetic and pharmacological inhibition of the kinases and phosphatases halted or mis-targeted trafficking vesicles[20,21]. Recent advanced live cell imaging showed that acute depletion of PtdIns(3)P from endosomal or phagosomal membranes results in a loss of specificity resembling immature compartments[22,23].

Generally, our knowledge about the functioning of zip code molecules is largely based on genetic perturbation approaches in intact cells, complemented with the study of budding and fusion under cell-free conditions using membrane fractions isolated from cells or tissues. These studies identified many molecules related to specific targeting and succeeded in reconstitution of vesicle fusion having specificity[1,24,25]. Due to the complexity of these systems in living cells it has been difficult to decipher how exactly different classes of zip-code molecules interact with each other, how they orchestrate the effector protein complexes and how they cooperate with membrane-resident components such as the SNARE fusion machinery. To address these issues, we have recently employed a new approach in which we introduced artificial vesicles with a precisely defined composition into living cells and then monitored their interactions with the cellular trafficking machineries. Surprisingly, artificial vesicles that were devoid of PtdInsP$_x$ and GTPases and that contained SNAREs suggested to function either in early or late endosomal fusion, respectively, as the only proteins (termed "early endosomal", and "late endosomal" SNAREs), fused within minutes preferentially (albeit not exclusively) with their endogenous counterparts. Targeting specificity was dependent on the recruitment of cytosolic proteins (e.g. tethering proteins) that specifically recognize the N-terminal domains of a SNARE or SNARE complexes, rather than on the specificity of the SNARE motifs required for fusion[26].

The question then arises how SNARE-dependent targeting cooperates with the canonical PtdInsP$_x$ and Rab targeting systems. Indeed, our data revealed that targeting specificity of SNARE liposomes was lower than that of isolated endosomal vesicles injected in parallel[27], suggesting that additional factors modulate the SNARE-dependent targeting. To shed light on this issue, we have now extended this approach and prepared liposomes containing not only endosomal SNAREs but also the Rab GTPases (Rab5, and Rab7) and/or the phosphoinositides (PtdIns(3)P, PtdIns(5)P, and PtdIns(3,5)P2) at various concentrations and examined their targeting after microinjection. Our results reveal a complex interplay between the targeting factors, which results in the recruitment of effector proteins that ultimately control the destination of the trafficking vesicle.

## Results
### Multiple sets of SNARE proteins do not interfere with each other in targeting
At the beginning we investigated whether different sets of SNARE proteins influence each other's targeting specificity when present on the same vesicle. As discussed above, we showed previously that proteoliposomes containing the SNAREs mediating homotypic fusion of early endosomes (syntaxin 13 (Qa), vti1a (Qb), syntaxin 6 (Qc), and

VAMP4 (R)[24,28,29], referred to as EE-SNAREs) preferably fused with early endosomal compartments. In contrast, proteoliposomes containing the SNAREs mediating fusion of late endosomes/lysosomes (syntaxin 7 (Qa), vti1b (Qb), syntaxin 8 (Qc), VAMP8 (R)[30], referred to as LE-SNAREs) preferably fused with late endosomes and lysosomes[26,27].

Here we reconstituted liposomes with both early and late endosomal SNAREs (8 different proteins in total), labeled them by incorporating 0.3% Rhodamine-phosphatidylethanolamine, micro-injected them into HeLa cells and analyzed their colocalization with a battery of marker proteins specific for endosomal subcompartments including APPL1 (adapter protein, phosphotyrosine interacting with PH domain and leucine zipper containing 1) for endocytic vesicles, EEA1 (early endosome antigen 1) for early endosomes, transferrin (Tfn) for early/recycling endosomes, MPR (mannose-6-phosphate receptor) for Golgi/endosome transport vesicles, LBPA (lysobisphosphatidic acid) for late endosomes, LAMP1 (lysosomal-associated membrane protein 1) for lysosomes, and Golgin97 for the *trans*-Golgi network (Supplementary Fig. S1a). As in our previous studies, colocalization was quantified by measuring the minimum distance of every injected liposome to the closest particle five min after injection. Liposomes localized within 100 nm from the organelle marker were defined as "co-localized", which indicates fusion with the target organelle[26,27]. Protein-free liposomes with a "basic" lipid composition (PSPC: phosphatidylcholine (PC, 79.7%), phosphatidylserine (PS, 20%), and Rhodamine-labeled phosphatidylethanolamine (Rh-PE, 0.3%), average diameter 100 nm) were used as control. Efficient reconstitution of proteins in the membrane was confirmed by analyzing the liposomes on flotation gradients (Supplementary Fig. S1b).

As shown in Fig. 1a and Supplementary Fig. S1c, liposomes containing both EE- and LE-SNAREs co-localized with both EE and LE markers. Generally, their distribution can be best described as a sum of the individual EE-SNARE and LE-SNARE liposomes, with only minor exceptions (lack of APPL1 targeting, and enhancement of targeting to the MPR compartment). These results suggest that multiple sets of SNAREs, when present in the same membrane, do not interfere with each other with respect to targeting and fusion.

### Phosphatidylinositol (3) phosphate and GTP-Rab5 control targeting of EE-SNARE-liposomes to early endosomal compartments
As outlined in the introduction, both phosphoinositides and Rab-GTPases are considered as essential components for defining the identity of a trafficking vesicle. Accordingly, early endosomes are characterized by the presence of phosphatidylinositol (3) phosphate (PtdIns(3)P), and the GTPase Rab5[11,31]. Therefore, we tested in which way the incorporation of these two signaling molecules influences targeting specificity of injected liposomes containing early endosomal SNAREs (EE-SNARE liposomes).

In the first set of experiments, we injected EE-SNARE liposomes containing increasing concentrations of PtdIns(3)P, which is known to be required for efficient trafficking of early endosomes[32]. Control experiments were carried out to ensure that both SNARE proteins and PtdIns(3)P were incorporated into the liposomes (Supplementary Fig. S2a-b). Note that in this and the following experiments we adjusted the composition of the membrane lipids to resemble that determined previously for early endosomes (31% PC, 14% phosphatidylethanolamine, 3% PS, 11% sphingomyelin, 40% cholesterol, 1% phosphatidylinositol)[33,34]. In the absence of PtdIns(3)P, colocalization with early endosomal markers (particularly with EEA1) appeared to be even more pronounced than with the simpler lipid mix used before (compare 0% PtdIns(3)P in Fig. 2a with 4-EE SNARE PL in Fig. 1a, see also reference[26]). Surprisingly, increasing concentrations of PtdIns(3)P strongly decreased colocalization, with no significant colocalization

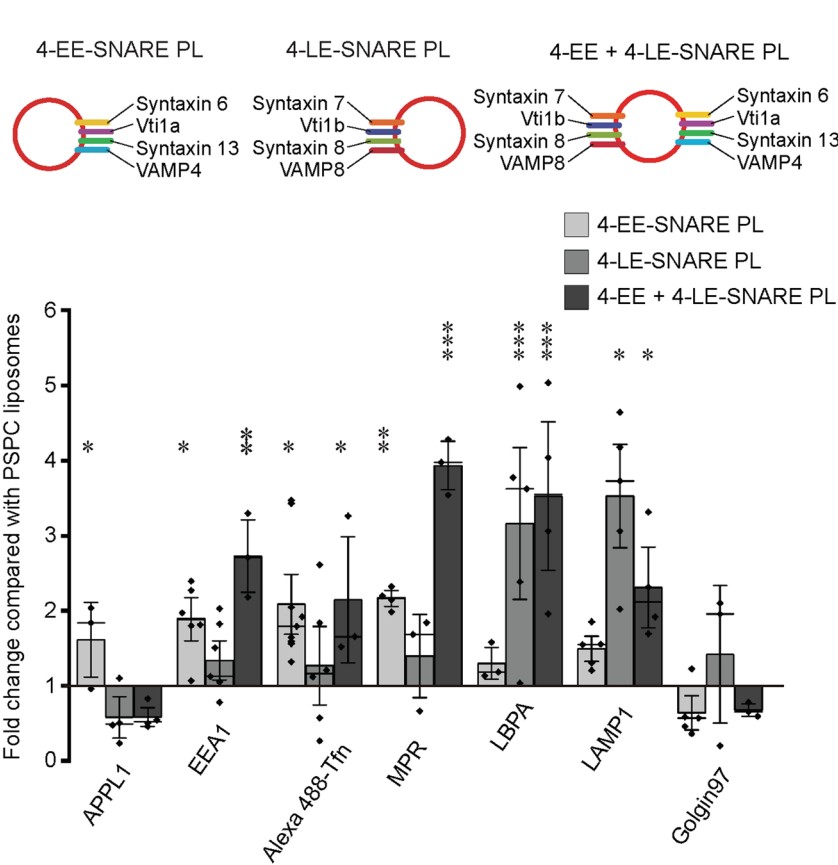

**Fig. 1 | SNARE proteins do not restrict targeting destination.** Colocalization between injected proteoliposomes (PLs) reconstituted with four early endosomal (EE) SNAREs (including syntaxin 13, vti1a, syntaxin 6, and VAMP4) (4-EE-SNARE PL) and/or late endosomal SNAREs (syntaxin 7, vti1b, syntaxin 8, VAMP) (4-LE-SNARE PL) and endogenous markers. In these experiments, all values were normalized to the degree of colocalization observed in control injections using protein-free liposomes containing a corresponding lipid composition with no further additions (see methods). The data show mean values ± SEM of 3–9 independent experiments, with each individual measurement being represented by a black dot. At least 100 injected vesicles were analyzed for the colocalization with each organelle marker in each experiment. Stars indicating significance: *$P < 0.05$, **$P < 0.01$, ***$P < 0.001$, all determined by 1-way ANOVA with the Tukey multiple comparison test. Source data are provided as a Source Data file.

with any of the endosomal markers being observable at 1% PtdIns(3)P. In contrast, colocalization with MPR positive vesicles was not inhibited (Fig. 2a and Supplementary Fig. S2c). The effects of PtdIns(3)P are specific since no effect on SNARE-dependent colocalization was observed when 1% PtdIns(5)P was incorporated instead of PtdIns(3)P (green bars in Fig. 2a). Control experiments revealed no significant colocalization with any of the organelle markers in the absence of SNARE proteins (Supplementary Fig. S2d).

The colocalization pattern observed above of 4-EE-SNAREs and 1% PtdIns(3)P liposomes is reminiscent of liposomes containing only syntaxin 6 in the absence of PtdIns(3)P described previously[26]. In that study we demonstrated that this targeting is mediated by the recruitment of Vps51, a component of the GARP tethering complex, to syntaxin 6[26]. We therefore asked whether the block by 1% PtdIns(3)P of EE-targeting may leave syntaxin 6 unaffected, allowing the protein to recruit Vps51 and to target the liposomes to the MPR-compartments. This seems to be the case since in the presence of 1% PtdIns(3)P the targeting of liposomes containing only syntaxin 6 was very similar to that of liposomes containing 4-EE-SNAREs (Fig. 2b). Similarly, the recruitment of Vps51 onto the syntaxin 6 liposomes was not affected by the presence of 1% PtdIns(3)P (Supplementary Fig. 2e), suggesting that the targeting of syntaxin 6 liposomes to the MPR-compartment is PtdIns(3)P independents.

In the next experiments, we examined whether inclusion of GTP-Rab5 that is known to be required for the fusion of early endosomes affects targeting of EE-SNARE liposomes. To this end, we purified prenylated versions of an active (GTPase-deficient) mutant (Rab5(Q79L)) and wt-Rab5 (converted into GDP-binding form in vitro) (see reference[35,36], Supplementary Fig. S3a) and incorporated them into liposomes (Supplementary Fig. S2b and Supplementary Fig. S3b). The concentration of Rab5 protein in liposomes was comparable to that of endogenous endosomes (calibrated by comparing the intensities of Rab5 immunofluorescence signals of injected liposomes with those of endogenous endosomes, Supplementary Fig. S3c, d). To our surprise, incorporation of the GTPase deficient Rab5 mutant selectively inhibited targeting of 4-EE-SNARE liposomes to early endosomes but not to the APPL1 or the MPR-compartments (Fig. 2c, Supplementary Fig. 3e). No such inhibition was observed when an active mutant of the late endosomal/lysosomal Rab7 was reconstituted instead (Rab7(Q67L), see Supplementary Fig. S3f for characterization), but lysosomal targeting was enhanced (Fig. 2c). We also tested the behavior of liposomes containing only syntaxin 6 in the presence of Rab5 which resulted in a pattern very similar to that of 4-EE- SNARE liposomes (Fig. 2d), suggesting that, as described above for PtdIns(3)P, targeting by syntaxin 6-Vps51 interaction is not affected by Rab5. In the absence of SNAREs, Rab5 did not cause colocalization with any of the endosomal markers (Supplementary Fig. S3g). Taken together, these findings show that the presence of either GTP-Rab5 or increasing concentrations of PtdIns(3)P selectively block EE-SNARE-dependent targeting of liposomes to early endosomes, with a conspicuous exception of syntaxin 6/Vps51 mediated targeting that does not appear to be affected under either condition.

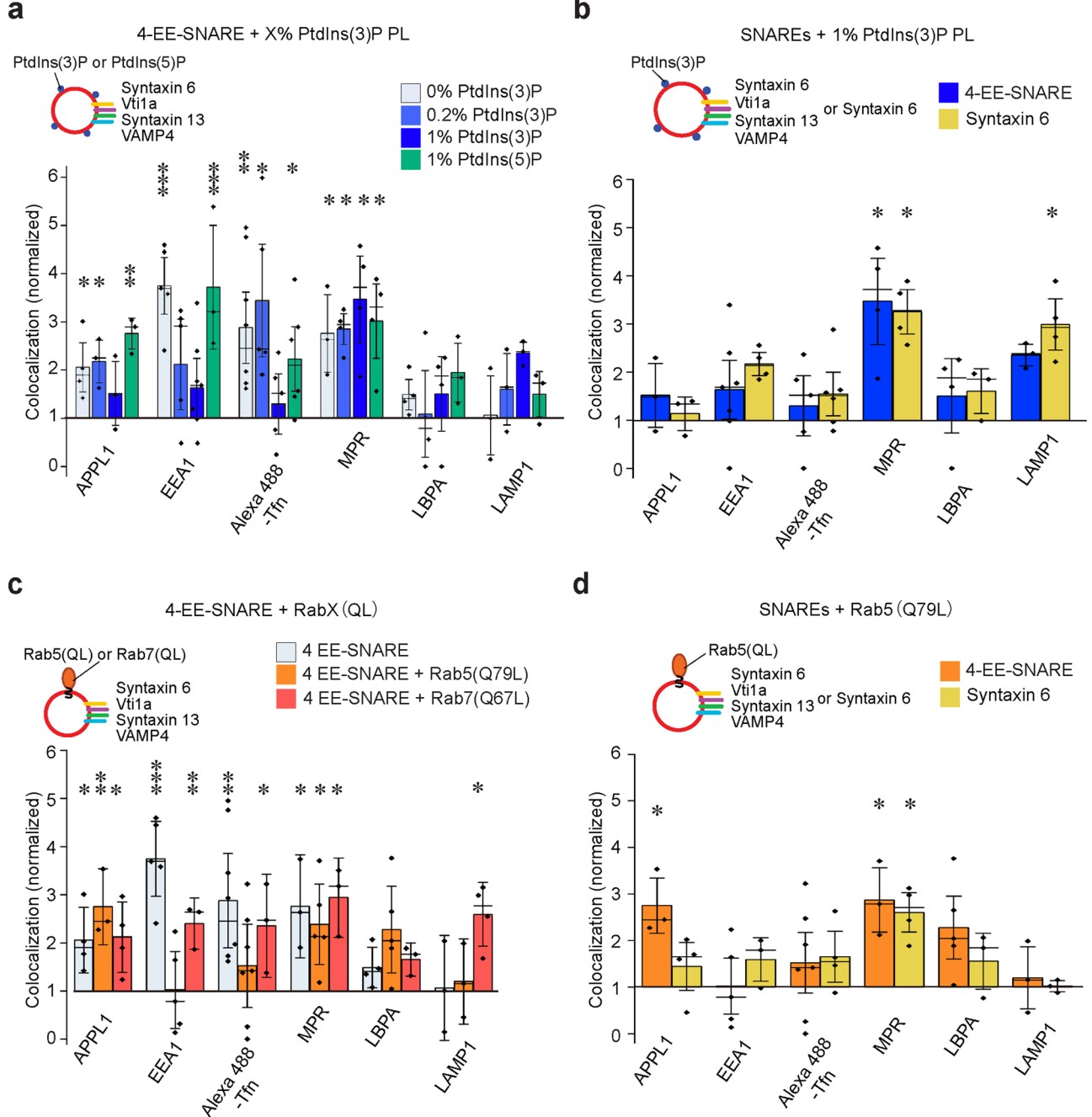

**Fig. 2 | Increasing concentrations of the membrane lipid PtdIns(3)P or GTP-Rab5 block targeting of EE-SNARE liposomes to early endosomes.**
**a** Colocalization between organellar markers and injected liposomes containing the 4-EE-SNAREs and PtdIns(3)P or, as control, PtdIns(5)P. Increasing concentrations of PtdIns(3)P in the liposome membrane selectively inhibit targeting of EE-SNARE PLs to endogenous early endosomes. **b** Colocalization between injected 1% PtdIns(3)P liposomes containing either four early endosomal SNAREs (4-EE-SNARE, blue) or syntaxin 6, respectively, and endogenous markers. Note that the data for 4-EE-SNAREs (blue) are the same as shown in Fig. 2a and included here for easier comparison. **c** Colocalization between organellar markers and injected liposomes containing 4 EE-SNAREs and prenylated constitutively active Rab5(Q79L) or prenylated constitutively active Rab7(Q67L). Active Rab5, but not active Rab7 selectively inhibits targeting of the liposomes to early endosomes. Instead, Rab7(Q67L),

but not Rab5(Q79L), induced targeting to LAMP1-positive lysosomes. Note that the data for 4-EE-SNAREs (light blue) are the same as shown in Fig. 2a and included here for easier comparison. **d** Colocalization between organellar markers and injected liposomes reconstituted with Rab5(Q79L) and either all four EE-SNAREs or only syntaxin 6. Inhibition of 4-EE-SNAREs dependent targeting to early endosomes by Rab5(Q79L) was also observed by liposomes with syntaxin6 and Rab5(Q79L). In all panels shown in this figure, the data represent mean values ± SEM of 3–7 independent experiments, with each individual measurement being represented by a black dot. At least 100 injected vesicles were analyzed for the colocalization with each organelle marker in each experiment. Stars indicate significance: $*P < 0.05$, $**P < 0.01$, $***P < 0.001$, all determined by 1-way ANOVA with the Tukey multiple comparison test. Source data are provided as a Source Data file.

It is well established that phosphoinositides operate in conjunction with Rab GTPases, with the binding of several Rab5-effectors required for early endosome fusion being enhanced if PtdIns(3)P is present ("coincidence detection"[18,31]). Therefore, we investigated targeting of

EE-SNARE liposomes in the presence of both GTP-Rab5 and PtdIns(3)P. Note that in these and the following experiments we used only the three Q-SNAREs (syntaxin 6, syntaxin 13, and Vti1a) for reconstitution because co-reconstitution of all SNAREs with Rab proteins and 1% PtdIns(3)P

resulted in liposome aggregation. As shown previously, liposomes containing these three Q-EE-SNAREs showed targeting almost identical to that of liposomes containing the full set of EE-SNAREs[26].

First, we injected EE-SNARE liposomes containing Rab5(Q79L) and 1% PtdIns(3)P. Again, no colocalization with any of the early endosomal markers was observable (Fig. 3a). Surprisingly, however, colocalization with late endosomal and lysosomal markers was markedly enhanced. This targeting requires the active GTP-form of Rab5 because it is only apparent when the GTP-preferring variant (Rab5(Q79L)) is used (Fig. 3a) and when the EE-SNAREs are present (Supplementary Fig. S4a). To explain this unexpected result, we considered previous reports showing that the PtdIns(3)P concentration is directly related to endosome maturation, with higher concentrations promoting the transition from early to late endosomes[37,38]. We therefore asked whether in the presence of active Rab5 targeting specificity may be regulated by the concentration of PtdIns(3)P. To this end, we repeated the experiment using proteoliposomes with lower PtdIns(3)P concentrations (0.05% and 0.2%, respectively). Indeed, at these lower PtdIns(3)P concentrations a high degree of colocalization with EEA1 but not with Transferrin was observed (Fig. 3b and Supplementary Fig. S4b). To explore this discrepancy further, the cells were incubated with fluorescence labeled epidermal growth factor (EGF) as a marker for the pathway leading from early endosomes to late endosomes, which differs from transferrin that is recycled back to the plasma membrane via recycling endosomes[39]. We also included Rab11 as marker for recycling endosomes. At low PtdIns(3)P concentrations, colocalization of the injected proteoliposomes with EGF-positive vesicles was enhanced whereas no significant colocalization with Rab11 was detectable. In contrast, enhanced targeting to the LBPA and LAMP1 compartment was not detectable (Fig. 3b and Supplementary Fig. S4b). Note that the colocalization with MPR (which shuttles between the Golgi and endosomes) was not affected by the presence of PtdIns(3)P, regardless of its concentration.

Several control experiments were carried out to substantiate these results. First, we replaced Rab5(Q79L) with the GTPase-deficient mutant of Rab7 described above (Rab7(Q67L), Fig. 3c). Unlike Rab5(Q79L), Rab7(Q67L) failed to enhance colocalization with the EEA1 compartment of liposomes containing 0.2% PtdIns(3)P, confirming that the targeting to the EEA1 compartment is specific for Rab5. In contrast, the GTPase deficient variants of both Rab proteins enhanced targeting to lysosomes when the membrane contained 1% PtdIns(3)P (Fig. 3c). Next, we examined which of the endosomal Q-SNAREs are needed for the targeting observed above. Omitting vti1a from these liposomes had no influence on the targeting described above whereas liposomes containing only syntaxin 6 in the presence of Rab5(Q79L) and 0.2% PtdIns(3)P are mainly targeted to the MPR-and LBPA compartments (Fig. 3d).

In summary, the data show that in the presence of active Rab5 on EE-SNARE liposomes, increasing PtdIns(3)P concentrations result in a gradual shift in liposome targeting and subsequent fusion from early to late endosomal compartments. In the absence of PtdIns(3)P, liposomes are targeted to endocytic vesicles (APPL1 compartment) but not to any of the early endosomal compartments. When the PtdIns(3)P is increased (as it happens during maturation) targeting to early endosomes becomes dominant while targeting to the APPL1-compartment is reduced. The fact that no colocalization is observed with recycling endosomes (in contrast to EE-SNARE liposomes lacking either Rab5 or PtdIns(3)P) suggests a "refinement" of the rather broad SNARE-dependent targeting. When the PtdIns(3)P concentration increases further, no targeting to any of the early endosomal compartments is observable, with the vesicles predominantly fusing with the late endosomal and even lysosomal compartments.

## Rabenosyn-5 inhibits targeting to early endosomes at high PtdIns(3)P concentrations

The data discussed so far show that either active Rab5 or increasing concentrations of PtdIns(3)P in the liposome membrane inhibit targeting and fusion of EE-SNARE liposomes with endogenous endosomes. The block is only relieved when active Rab5 is combined with a low, permissive concentration of PtdIns(3)P, and this dis-inhibition is specific only for fusion with early but not with recycling endosomes labeled with transferrin.

What are the mechanisms underlying this fine-tuning? It is well established that fusion between early endosomes depends on a set of tethering factors that possess PtdIns(3)P binding domains and that can bind to Rab5 or additional Rab proteins operating in the endolysosomal system (coincidence detection, see above). Indeed, in a de-novo reconstitution of SNARE-containing liposomes maximal fusion was only observed when these proteins were included[24]. We therefore examined whether one or several of these factors may also be involved in the inhibition of SNARE-dependent targeting at high PtdIns(3)P concentrations.

To this end, we isolated endosomes from HeLa cells pre-labeled with fluorescent Transferrin and measured fusion between these endosomes and liposomes in vitro using a previously described particle-counting assay[40,41]. As shown in Fig. 4a, inclusion of 1% PtdIns(3)P into the membrane of EE-SNARE liposomes inhibited fusion with Transferrin-labeled endosomes, thus reproducing the results obtained by microinjection of the liposomes (compared with Fig. 2a). We then examined whether one of the following PtdIns(3)P binding proteins contributes to the observed inhibition: EEA1, Hrs, Rabenosyn-5, and Rabankyrin-5. Cytosol extracts were prepared from which each of these proteins was depleted by immunoprecipitation and then used in the in vitro fusion assays (Fig. 4b). Control experiments in which fusion between endogenous Transferrin-labeled endosomes was measured showed that removal of each protein significantly impaired fusion, in agreement with previous reports (Fig. 4c)[42–45]. We then used these depleted cytosols to determine whether they influence fusion between 4-EE-SNARE liposomes containing inhibitory 1% PtdIns(3)P and Transferrin-positive endosomes. Strikingly, Rabenosyn-5 depleted cytosol, but not any of the others, rescued the block and restored fusion of these liposomes with Transferrin-labeled endosomes (Fig. 4d). To gain further insight into the underlying regulatory network, we included active Rab5 into the membrane and monitored fusion with EEA1 positive endosomes at different PtdIns(3)P concentrations. While not reaching high significance, the results reveal trends confirming that while Rabenosyn-5 supports fusion with EEA1-endosomes at low PtdIns(3)P concentrations it reduces fusion at high concentrations (Fig. 4e). Note that the amount of Rabenosyn-5 recruited to the injected 1% PtdIns(3)P liposomes is comparable to that of endogenous Transferrin-positive endosomes (Fig. 4f, g). Rabenosyn-5 effectively binds to the membrane even in the absence of Rab5 (Fig. 4f, g). The recruited levels of Rabenosyn-5 on 0.2% and 1% PtdIns(3)P was similar (Fig. 4h), suggesting that the opposing effects of Rabenosyn-5 on these liposomes do not depend on the protein level on the membrane.

## Conversion of PtdIns(3)P to PtdIns(3.5)P2 changes targeting from early endosomes to late endosomes

We next investigated why at high PtdIns(3)P concentrations, EE-SNARE liposomes are selectively targeted to the late endosomal/lysosomal compartments despite the presence of constitutively active Rab5 (Fig. 3a). It is well established that maturation of early into late endosomes is associated both with an exchange of Rab5 for Rab7, which is mediated by the Mon1-Ccz1 complex[46,47] and a switch from PtdIns(3)P to PtdIns(3.5)P, which is mediated by the recruitment of the PtdIns(3)P-kinase PIKfyve. PIKfyve directly binds to PtdIns(3)P via FYVE domain (see diagram in Fig. 5a). We therefore asked whether either one or both of these exchanges may occur on the injected liposomes containing EE-SNAREs, active Rab5, and 1% PtdIns(3)P, thus explaining the change in targeting specificity. First, we tested whether Rab5 is exchanged for

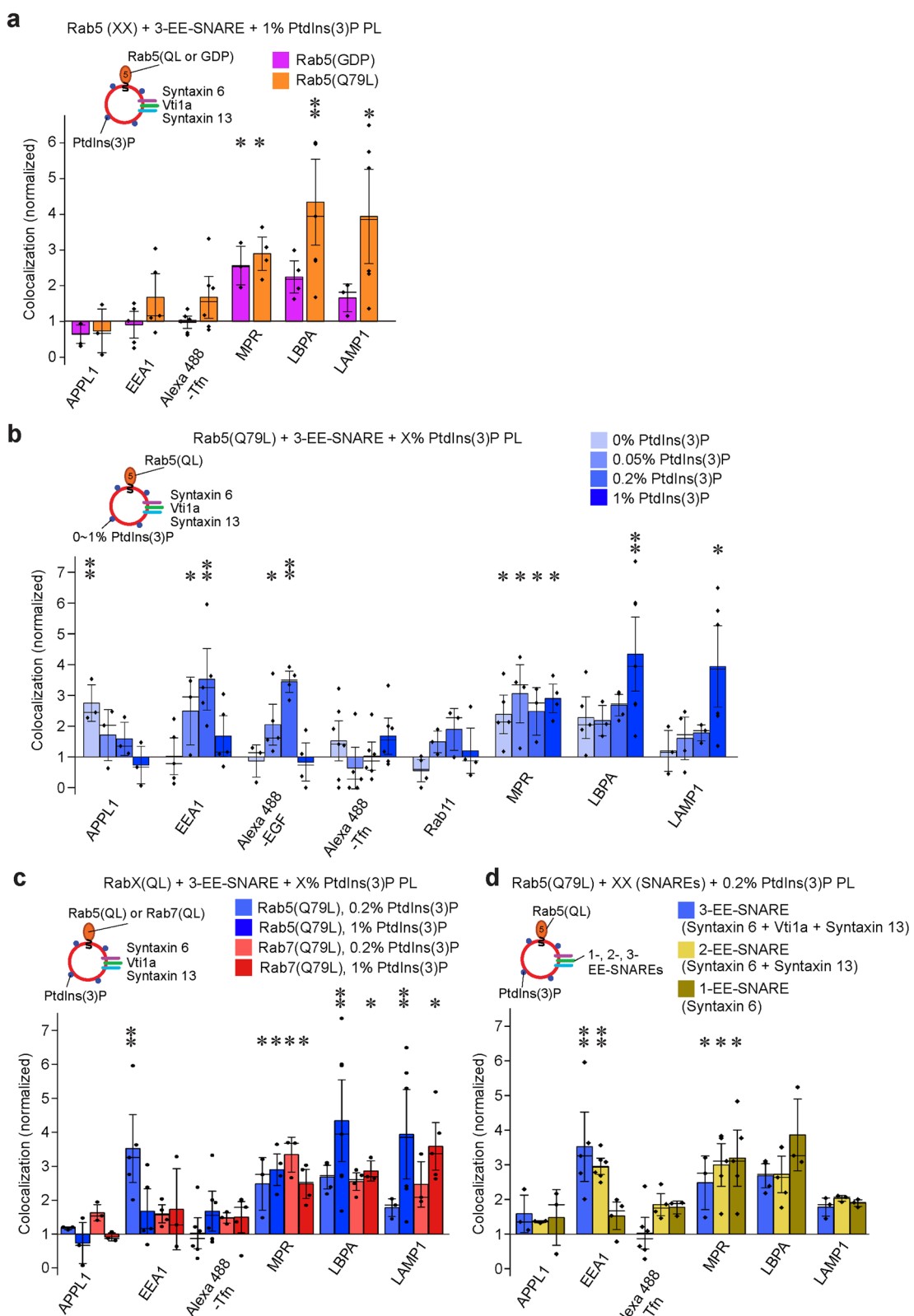

Rab7 using several independent approaches. However, we were unable to detect recruitment of Rab7, Mon1, or RILP (Rab Interacting Lysosomal Protein, a Rab7 effector protein) to the liposomes in vitro even when cytosolic extracts from cells overexpressing GFP-Rab7 or GFP-Mon1 were used (Supplementary Fig. 5a). In contrast, endogenous PIKfyve was recruited to the proteoliposomes in a PtdIns(3)P concentration-dependent manner (Fig. 5b). Selective recruitment of

PIKfyve (but not Rab7) was confirmed when liposomes incubated with extracts containing GFP-tagged versions of the proteins and then analyzed by fluorescence microscopy (Supplementary Fig. S5b, c).

Next we tested whether conversion of PtdIns(3)P to PtdIns(3,5)P2 also occurs in the cell after liposome injection. To this end, we expressed a reporter for PtdIns(3,5)P2 in HeLa cells, consisting of two copies of the lipid binding domain of TRPML fused to EGFP (GFP-ML1N*2) shown

**Fig. 3 | In the presence of GTP-Rab5, PtdIns(3)P fine-tunes targeting of EE-SNARE liposomes in a concentration-dependent manner. a** Colocalization between organellar markers and injected liposomes containing the three EE Q-SNAREs, prenylated Rab5 (Q79L) or GDP (wt)-Rab5, and 1% PtdIns(3)P. No targeting to early endosomes is observable whereas targeting to late endosomes/lysosomes is markedly enhanced but only when the GTP-form of Rab5 is present.
**b** Colocalization between organellar markers and injected liposomes containing the three EE-Q-SNAREs, prenylated Rab5 (Q79L), and increasing concentrations of PtdIns(3)P. Note that the control liposomes contained, in addition, the corresponding concentrations of PtdIns(3)P. At low concentrations of PtdIns(3)P targeting to early but not to recycling endosomes is enhanced. At high PtdIns(3)P concentrations, targeting to early endosomal compartments is inhibited whereas instead targeting to late endosomes/lysosomes is enhanced. Note that the data for 1% PtdIns(3)P are the same as shown in Fig. 3a and included here for easier comparison. **c** Comparison of the colocalizations between organellar markers and injected liposomes containing either Rab5(Q79L) or Rab7(Q67L) at low (0.2%) and high (1%) PtdIns(3)P concentrations (the Rab5 (Q79L) data are from Fig. 3b and shown for comparison). Replacing GTP-Rab5 with GTP-Rab7 abolishes targeting to the EEA1 compartment at low PtdIns(3)P concentrations but resembles GTP-Rab5 in enhancing targeting to lysosomes at high PtdIns(3)P concentrations.
**d** Colocalization between organellar markers and injected liposomes containing Rab5(Q79L), 0.2% PtdIns(3)P, and either all three EE-Q-SNAREs, syntaxin 13 and syntaxin 6, or only syntaxin 6. Omission of Vti1a resulted in targeting very similar to the 3 EE-Q-SNARE liposomes, but when only syntaxin 6 was present, a selective loss of EEA1 targeting was observed. Data of 3-EE-SNARE liposomes are from Fig. 3b for easier comparison. In all panels shown in this figure, the data represent mean values ± SEM of 3–6 independent experiments, with each individual measurement being represented by a black dot. at least 100 injected vesicles were analyzed for the colocalization with each organelle marker in each experiment. Stars indicate significance: *$P < 0.05$, **$P < 0.01$, ***$P < 0.001$, all determined by 1-way ANOVA with the Tukey multiple comparison test. Source data are provided as a Source Data file.

previously to be specific for this phosphoinositide[48]. After confirming that this reporter specifically binds to PtdIns(3.5)P2-containing liposomes (Supplementary Fig. S5d), we incubated proteoliposomes containing increasing PtdIns(3)P concentrations with cytosolic extracts from HeLa cells expressing the reporter protein and measured binding by flotation gradients. Indeed, GFP-ML1N*2 was recruited to the membrane in a PtdIns(3)P concentration- and incubation time-dependent manner (Fig. 5b and Supplementary Fig. S5e). Interestingly, active-Rab5 appeared to enhance the generation of PtdIns(3.5)P2 (Fig. 5b). As expected, treatment of PIKfyve inhibitor (YM-201636) prevented the recruitment of the probe (Supplementary Fig. S5f).

Taken together, the data presented above show that liposomes containing EE-SNAREs, active Rab5, and 1% PtdIns(3)P generate PtdIns(3.5)P2 by recruiting PIKfyve but do not exchange Rab5 for Rab7. In the final set of experiments, we therefore investigated whether this switch is responsible for redirecting the liposomes to late endosomes/lysosomes despite the presence of early endosomal SNAREs and active Rab5. First, we tested whether downregulation of Rab7 expression would have any effect on the targeting specificity which was not the case (Supplementary Fig. S5g). Next, we incubated injected cells with YM-201636 to prevent PtdIns conversion, resulting in reduced targeting of the liposomes to late endosomes/lysosomes (Fig. 5c and Supplementary Fig. S6a). Finally, we prepared liposomes containing 0.2% PtdIns(3.5)P2 containing either EE-SNAREs and active Rab5, only EE-SNAREs, or no protein and injected them into HeLa cells. Whereas protein-free liposomes resulted in no significant colocalization with any of the organelle markers (Supplementary Fig. S5h), EE-SNARE-liposomes containing 0.2% PtdIns(3.5)P2 were highly colocalized with lysosomal markers as well as early endosomal markers and MPR-containing vesicles (Fig. 5d and Supplementary Fig. S6b). Further inclusion of Rab5 (Q79L) showed a similar targeting pattern except for decreasing colocalization with EEA1-positive early endosomes (Fig. 5d and Supplementary Fig. S6b), suggesting that Rab5 activity is no longer necessary for lysosomal targeting of vesicles containing 0.2% PtdIns(3.5)P2. Taken together, the data show that high concentrations of PtdIns(3)P recruits PIKfyve to the vesicle membrane, followed by the generation of PtdIns(3.5)P2 which overrides the EE-SNAREs dependent targeting specificity and targets vesicles to lysosomes, even if active Rab5 is present.

We conclude that the interplay between the phosphoinositides and the GTPase Rab5 fine-tunes targeting specificity, most probably by selectively recruiting effectors in a manner that depends both on the activity of Rab5 and the concentration and type of phosphoinositides in the membrane (see Discussion).

## Discussion

Previously, we showed that artificial vesicles containing as few as 1–2 early endosomal SNAREs and a basic two-component mix of membrane lipids, when microinjected into living cells, target rather broadly to endogenous early and recycling endosomes where they dock and fuse[27]. In the present study, we have extended this approach by including various concentrations of PtdIns(3)P and/or the GTPase Rab5 into the liposome membrane, resulting in the preferential targeting to distinct endosomal subsets including promotion towards late endosomes at elevated PtdIns(3)P concentrations. Our data not only confirm previous studies dealing with the roles of Rab5 and phosphoinositides in endosomal targeting but uncover an unexpected cross-talk between competing targeting signals, suggesting that targeting is refined by a combination of inhibitory and permissive signals that dynamically interact with each other. A cartoon summarizing our main findings is shown in Fig. 6.

The trafficking routes to and from early (sorting) endosomes and the molecular machineries underlying the individual budding, docking, and fusion steps are among the best characterized in the secretory pathway. Using appropriate markers, endosomal trafficking can be easily monitored in living cells, with an ever-increasing toolbox being available for addressing individual steps of the pathway. Fusion between early endosomes (homotypic fusion) was among the first trafficking steps reconstituted in vitro using purified endosomes and soluble protein extracts, making the reaction accessible to biochemical manipulations[6,7]. Crucial factors required for remodeling, tethering, docking and fusion were identified, which culminated in the complete ab-initio reconstitution of early endosome fusion from purified components[24].

Introducing artificial vesicles with a defined composition into intact cells and monitoring their fate adds another tool to the study of vesicular traffic, forming a bridge between in cellulo manipulations and in vitro assays from purified or reconstituted components. It is obviously a drawback that remodeling of membrane lipids and recruitment of proteins is more difficult to control after microinjection than in an in vitro setup, but this is more than offset by the advantage that targeting of synthetic vesicles with a fully defined "starting" composition can be investigated in a completely native environment. Moreover, all measurements were carried out 5 min after completing injection, thus reducing possible artefacts due to "non-specific" remodeling.

In general, our data confirm and extend previous observations suggesting that an increase in the PtdIns(3)P concentration is tightly associated with (and may even control) the maturation of early to late endosomes. PtdIns(3)P appears to be absent from endosomal precursors (containing APPL1) but gradually increases upon maturation[22]. Acute depletion of PtdIns(3)P reverts endosomes from more mature to immature stages[22]. In yeast, PtdIns(3)P is first detectable in the membrane directly after endocytosis and then increases by orders of magnitude during progress from early (Vps21-positive) to late (Ypt7-positive) endosomes[37]. Indeed, we observe that at

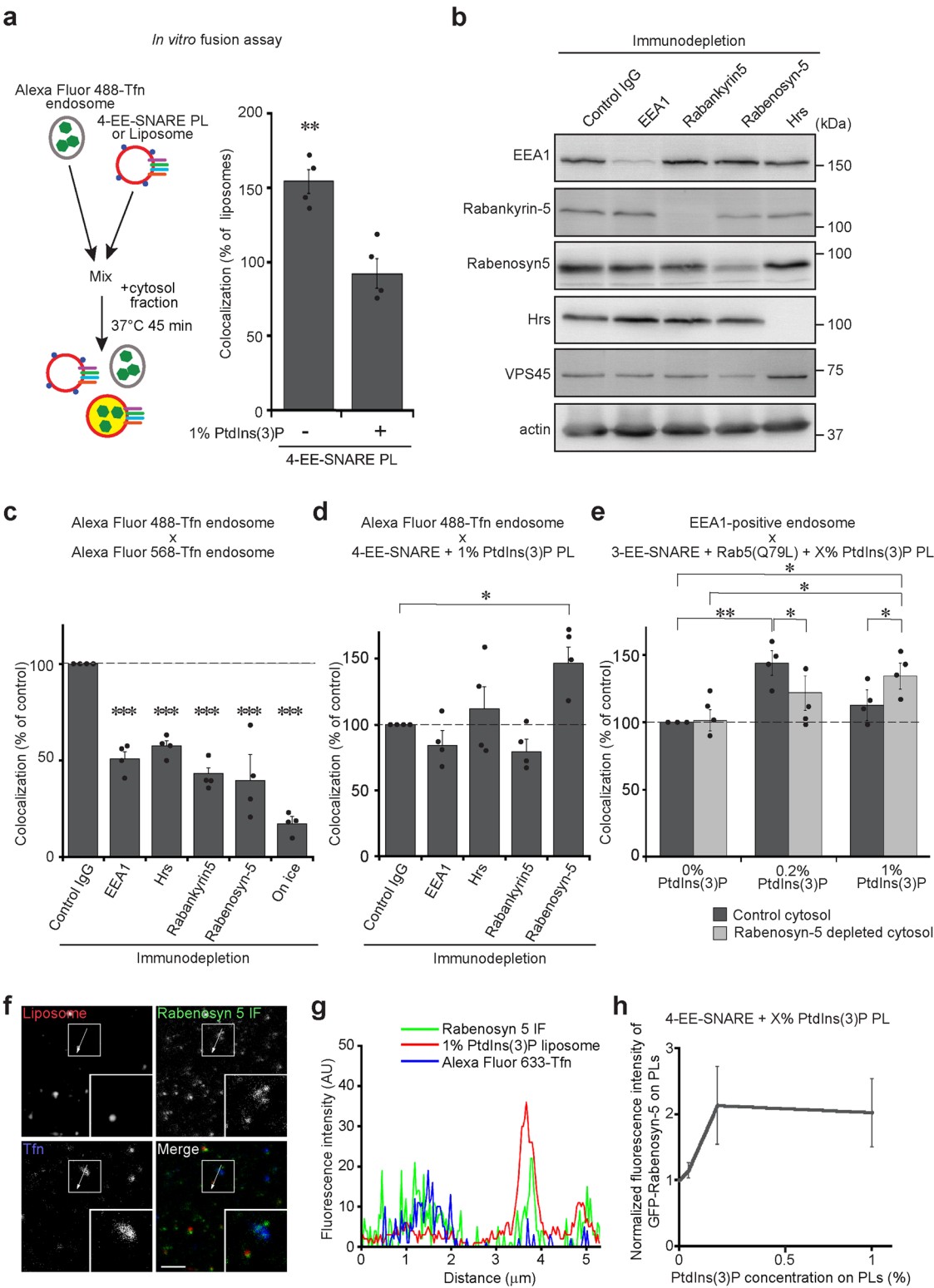

low PtdIns(3)P levels, Rab5 (Q79L), but not Rab7(Q67L), directs injected liposomes to a subset of endosomes that is clearly distinct from those involved in Transferrin recycling but rather part of the degradative pathway, marked here by EGF. This agrees with previous work showing that Rab5 mainly functions in this pathway[49] whereas recycling to the plasma membrane is governed by a transition from Rab5 to Rab4 and Rab11[9]. Increasing PtdIns(3)P concentrations shift the targeting preference from early to late endosomes, which is dependent on the recruitment of the PtdIns(3)P kinase PIKfyve that

generates PtdIns(3,5)P, again in agreement with previous studies[50] (see below).

Being able to introduce proteoliposomes with a precisely defined starting composition allowed for separately dissecting the roles of Rabs and phosphoinositides, resulting in several rather surprising observations. First, when EE-SNARE liposomes are supplemented either with PtdIns(3)P or GTP-Rab5, SNARE-dependent targeting to early endosomes is completely inhibited, with the block only being relieved when both PtdIns(3)P and GTP-Rab5 (Q79L) are present. Thus,

**Fig. 4 | Rabenosyn-5 contributes to inhibition of SNARE dependent targeting.**
**a** In vitro fusion assay between liposomes containing early endosomal SNAREs (4-EE-SNARE PL) and early endosomes, labeled by incubation with fluorescent transferrin. Values were normalized to the degree of colocalization observed in control protein-free liposomes. Inclusion of 1% PtdIns(3)P in the membrane of 4-EE-SNARE-PL inhibits fusion. The data show mean values ± SEM of 4 independent experiments. At least 100 liposomes were analyzed for the colocalization with Transferrin-positive endosomes in each experiment. **P < 0.01, determined by an unpaired two-tailed t-test. **b** Immunoblot showing effective immunodepletion of PtdIns(3)P binding proteins from HeLa cell-derived cytosol fractions. VPS45, which is a SM protein, co-depleted only in Rabenosyn-5 depleted cytosol. Actin was loading control. Same experiments were independently performed three times. **c** Depletion of the PtdIns(3)P binding proteins significantly inhibited homotypic fusion of endosomes in vitro. **d** In vitro fusion assay between 1% PtdIns(3)P liposomes containing early endosomal SNAREs (4-EE-SNARE PL) and early endosomes as in (**a**) using immunodepleted cytosol fractions. **e** In vitro fusion assay as in (**a**) using EE-SNARE liposomes reconstituted with Rab5(Q79L) and increasing concentrations of PtdIns(3)P. Note that in this experiment we used EEA1 as marker for early endosomes. The data in (**c–e**) show mean values ± SEM of 4 independent experiments. At least 100 liposomes were analyzed for the colocalization with Transferrin-positive endosomes in each experiment. *P < 0.05, **P < 0.01, ***P < 0.001, all determined by 1-way ANOVA with the Tukey multiple comparison test. **f** Comparison of Rabenosyn-5 protein levels between injected 1% PtdIns(3)P liposomes and endogenous Transferrin-positive endosomes in cells. To label early endosomes, Alexa Fluor 633-Transferrin was internalized, and then 1% PtdIns(3)P liposomes were microinjected followed by immunostaining (IF) for Rabenosyn-5. Inserts show a higher magnification of the area surrounded by white boxes. Same experiments were independently performed two times and images were captured from four different cells in each experiment. Scale bar, 5 μm. **g** An intensity plot of the line scan (a white line) in the pictures on (**f**). **h** Microscopy-based detection of GFP-Rabenosyn-5 recruitment to liposomes. The data show mean values ± SEM of 6 independent experiments. Source data are provided as a Source Data file.

constitutive SNARE-dependent targeting is fine-tuned, e.g. by preventing the recruitment of SNARE-specific tethering factors[26,27] or by blocking their activity.

Second, it is apparent that the conversion of PtdIns(3)P (at high concentrations) into PtdIns(3.5)P2 is dominant in targeting the liposomes towards late endosomes/lysosomes, being able to at least partially override the specificity conveyed by the EE-SNAREs and Rab5. Intriguingly, this switch requires active GTP-Rab5. Indeed, we observed that proteoliposomes containing active Rab5 effectively recruit PIKfyve and generate more PtdIns(3.5)P2, implying that the combination of Rab5 and high concentrations of PtdIns(3)P is required for the efficient recruitment of the enzyme. It remains to be explained why the switch from Rab5 to Rab7 that is known to be associated with endosomal maturation was not observed on our liposomes. Previously, the Mon1-Ccz1 complex was shown to be recruited to liposomes in a manner dependent on phospholipids and membrane-associated Rab5 proteins[51]. However, in our experiments we did not observe recruitment of endogenous Mon1-Ccz1 complex to injected liposomes, even in the presence of Rab5 and PtdIns(3)P. There are some possibilities to explain this discrepancy. First, cytosolic proteins may interfere with the ability of the injected liposomes to recruit the Mon1-Ccz1 protein complex. Second, overexpressed GFP-Mon1 may not be functional, i.e. it may not be able to form a dimer with Ccz1. Finally, we cannot exclude that other factors are needed for the recruitment of the Mon1-Ccz1 protein complex. For instance, the Mon1-Ccz1 complex was shown to bind to the HOPS complex and to Rabex5[46], with membrane recruitment being regulated by phosphorylation via Yck3 kinase[51]. Thus, while mechanistic details are still unclear, our data clearly reveal that PtdIns(3.5)P2 generated from PtdIns(3)P may play a much more critical role as a "zip code" for late endsomal/lysosomal targeting than previously assumed, and it will be important to identify and characterize additional PtdIns(3.5)P2 effector proteins that are responsible for this function. Note that once targeted to late endosomes, early endosomal SNAREs are capable of carrying out effective fusion, confirming that assembly of SNARE motifs driving fusion is rather promiscuous, particularly in the endosomal system[52,53].

Third, an unexpected additional role of Rabenosyn-5 was uncovered in preventing fusion with early endosomes at high PtdIns(3)P concentrations, which is different from the other effectors required for homotypic fusion at low PtdIns(3)P concentrations. Such dual role agrees with the fact that Rabenosyn-5 is an evolutionarily conserved multidomain protein with multiple functions in endosomal recycling and degradation[43,54,55]. In addition to its PtdIns(3)P binding fyve domain it possesses, among others at least three independent Rab-binding domains. Rabenosyn-5 not only interacts with multiple Rab proteins (Rab5, Rab4. Rab22, Rab24)[56], but also with SNARE-regulating SM proteins (Vps45 and VPS33B), EHD (Eps15 Homology Domain) Proteins (EHD1, EHD2, EHD3), and the multisubunit HOPS tethering factor[57–59]. Depletion or overexpression of Rabenosyn-5 results in multiple trafficking defects, mislocalizations and developmental aberrations[43,55,60,61].

In summary, our data reveal that membrane trafficking, despite its complexity, is highly robust and driven by stable self-assembly pathways, which allows for the integration of "foreign" vesicles with a minimal set of proteins (i.e. a single SNARE) and a simple lipid composition into existing trafficking pathway. Moreover, they reveal a complex interplay between the three types of "zip codes" (GTPases, phosphoinositides, and SNAREs) that each contribute to targeting by the recruitment of effectors and tethering complexes, either on their own or in a combinatorial manner. Evidently, shifts in the spectrum of effector proteins upon changes in the PtdIns(3)P concentration or in the conversion of PtdIns(3)P to PtdIns(3.5)P2 can change targeting specificity of vesicles, overriding targeting signals of Rabs and SNAREs, with combinatorial fine-tuning being responsible for sharpening the specificity of targeting. Here we have followed up two of these factors (Rabenosyn-5 and PIKfyve), documenting that our approach is suitable for dissecting the functional diversity of such effectors. Note that our approach can easily be extended, for instance for studying the result of "mismatches" between the targeting signals, by mis-directing vesicles to non-canonical membranes, or for investigating membrane recruitment to injected liposomes using fluorescently labeled proteins.

## Methods
### Materials
All phospholipids were obtained from Avanti Polar Lipids. Alexa Fluor 488-, Alexa Fluor 568-, or Alexa Fluor 633-conjugated Transferrin (T13342, T23365, T23362) and Alexa Fluor 488-conjugated EGF (E13345) were from Molecular Probes. YM-201636 was obtained from Santa Cruz Biotechnology. DAPI (4',6-diamidino-2-phenylindole) was from Sigma-Aldrich.

### DNA constructs
To generate pEGFP-C1-PIKfyve, a plasmid containing the cDNA encoding Human PIKfyve (#20584, Addgene) was used. The cDNA encoding dog Rab7A (WT) and mouse Mon1 were cloned into pEGFP-C1 vector. The mutated Rab7 (Q67L) construct was prepared by PCR-mediated site-directed mutagenesis using pEGFP-C1-Rab7 (WT). Syntaxin 13, vti1a, syntaxin 6, VAMP4, syntaxin 7, vti1b, syntaxin 8, and VAMP8, derived from *Rattus norvegicus*, were cloned into pET28 vectors (Merck Millipore) for bacterial protein purification[29,30]. Human Rab5A (WT and Q79L mutant) and dog Rab7A (Q67L) were cloned into pFastBac Dual vectors (Thermo Fischer Scientific) for insect cell expression and protein purification. GFP-2xFYVE and GFP-ML1N*2 plasmids were kindly provided by Dr. Harald Stenmark and Dr. Haoxing Xu, respectively. The shRNA vector targeting RAB7A (TRCN0000007996) was obtained from GE Dharmacon.

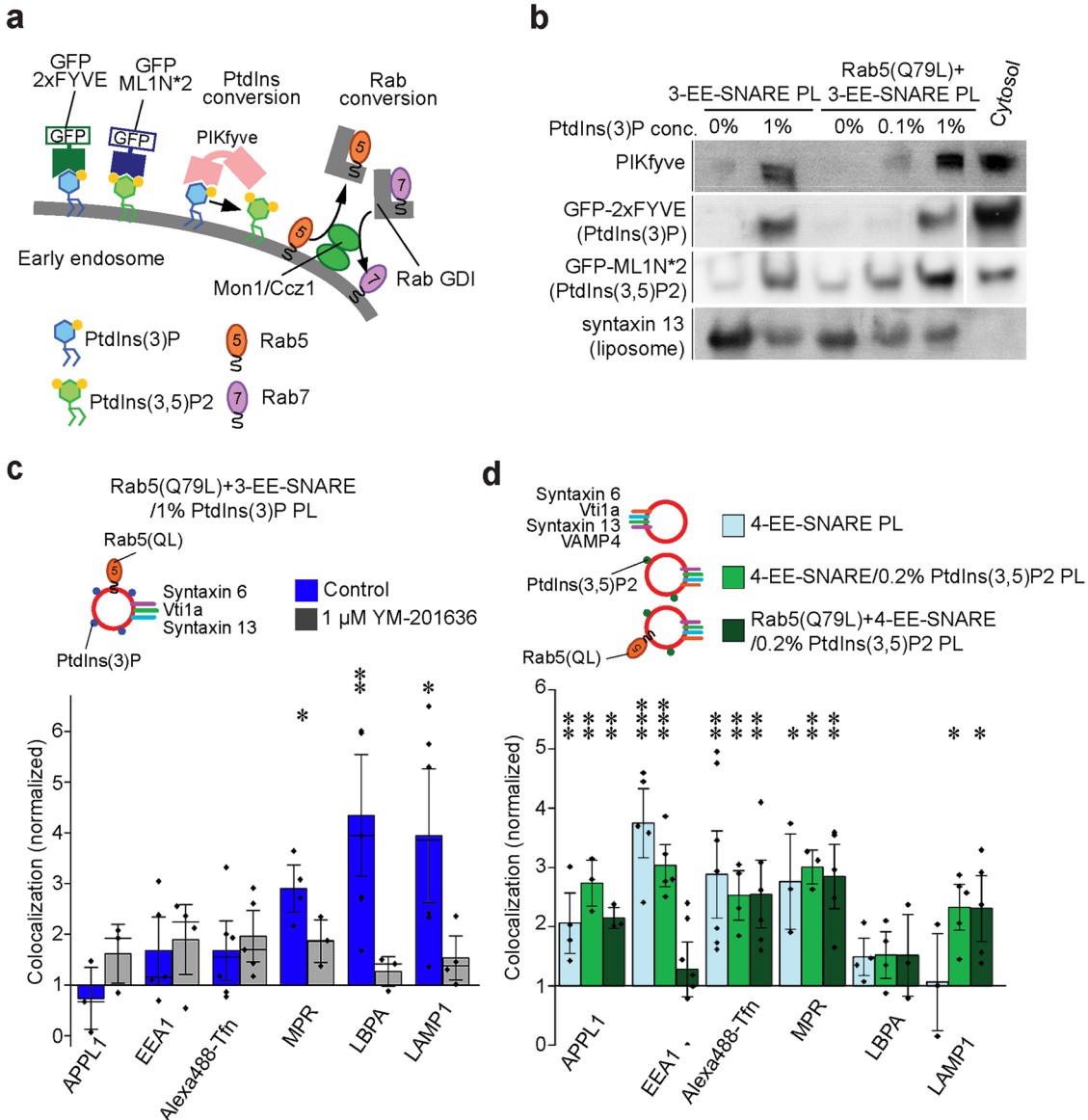

**Fig. 5 | Conversion of PtdIns(3)P to PtdIns(3.5)P2 changes targeting from early endosomes to late endosomes. a** Diagram showing the probes used for detecting PtdIns(3)P and PtdIns(3.5)P2, and mechanisms proposed for converting PtdIns(3)P to PtdIns(3.5)P2 and for exchanging Rab5 for Rab7. **b** Binding of PtdInsPx binding proteins to proteoliposomes containing EE-SNAREs or EE-SNAREs plus Rab5(Q79L) in the presence and absence of PtdIns(3)P. All liposomes contained 1% Biotin-PE. To detect PIKfyve recruitment, the proteoliposomes were incubated with cytosolic fractions for 30 min at 37 °C. The proteoliposomes were precipitated with Strep-tavidin beads and recruited PIKfyve was detected by immunoblotting. PtdIns(3)P or PtdIns(3.5)P2 on proteoliposomes were detected by EGFP-2xFYVE or GFP-ML1N*2 probes that were expressed in HeLa cells (incubation with cytosol from transfected cells), with binding being measured by flotation gradients. Syntaxin 13 indicates loading control. **c** Targeting specificity of liposomes containing three EE-SNAREs,

active Rab5, and 1% PtdIns(3)P in 1 μM YM-201636 treated cells. The inhibition of PtdIns(3.5)P2 generation prevents the targeting of the proteoliposomes to late endosomes/lysosomes. The data show mean values ± SEM of 3–6 independent experiments. At least 100 injected vesicles were analyzed for the colocalization with each organelle marker in each experiment. Stars indicating significance: *$P < 0.05$, **$P < 0.01$, all determined by 1-way ANOVA with the Tukey multiple comparison test. **d** PtdIns(3.5)P2 shifts targeting specificity of EE-SNARE PL to late endosomes/lysosomes. See text for details. The data show mean values ± SEM of 3–7 independent experiments. At least 100 injected vesicles were analyzed for the colocalization with each organelle marker in each experiment. Stars indicating significance: *$P < 0.05$, **$P < 0.01$, ***$P < 0.001$, all determined by 1-way ANOVA with the Tukey multiple comparison test. Source data are provided as a Source Data file.

## Antibodies

Primary antibodies used were obtained from the following companies: anti-APPL1 (3858) and anti-Rab7 (9367) from Cell Signaling; anti-EEA1 (612006) from BD Biosciences; anti-M6PR (ab2733), anti-RILP (ab140188) and anti-LAMP1 (ab24170) from Abcam; anti-PIKfyve (AF7885) from R&D systems; anti-LBPA (Z-PLBPA) from Echelon; anti-Golgin97 (A-21270), anti-Hrs (PA5-27491) and anti-Rab11 (71-5300) from Thermo Fischer Scientific; anti-Rabenosyn-5 (222218-1-AP) from Proteintech; anti-Rabankyrin5 (11321) and anti-VPS51 (HPA039650)

from Sigma-Aldrich; anti-GFP (132002), anti-β-actin (251003; 1:10,000 dilution for Western blotting), anti-Rab5 (108011), anti-Syntaxin 6 (110062), anti-Syntaxin 13 (110132), anti-Vti1a (165003), anti-VAMP4 (136002), anti-Syntaxin 7 (110072), anti-Syntaxin 8 (110083), anti-Vti1b (164002), and anti-VAMP8 (104302) from Synaptic Systems. For Western blotting, the appropriate primary antibodies were used at a dilution of 1:1000 (or as otherwise stated). Alexa Fluor 488-, Cy3-, or Cy5-conjugated goat anti-mouse (115–545–166, 115–165–146, 115–175–166) and goat anti-rabbit IgG (111–545–144, 111–165–144, 111–175–144) were

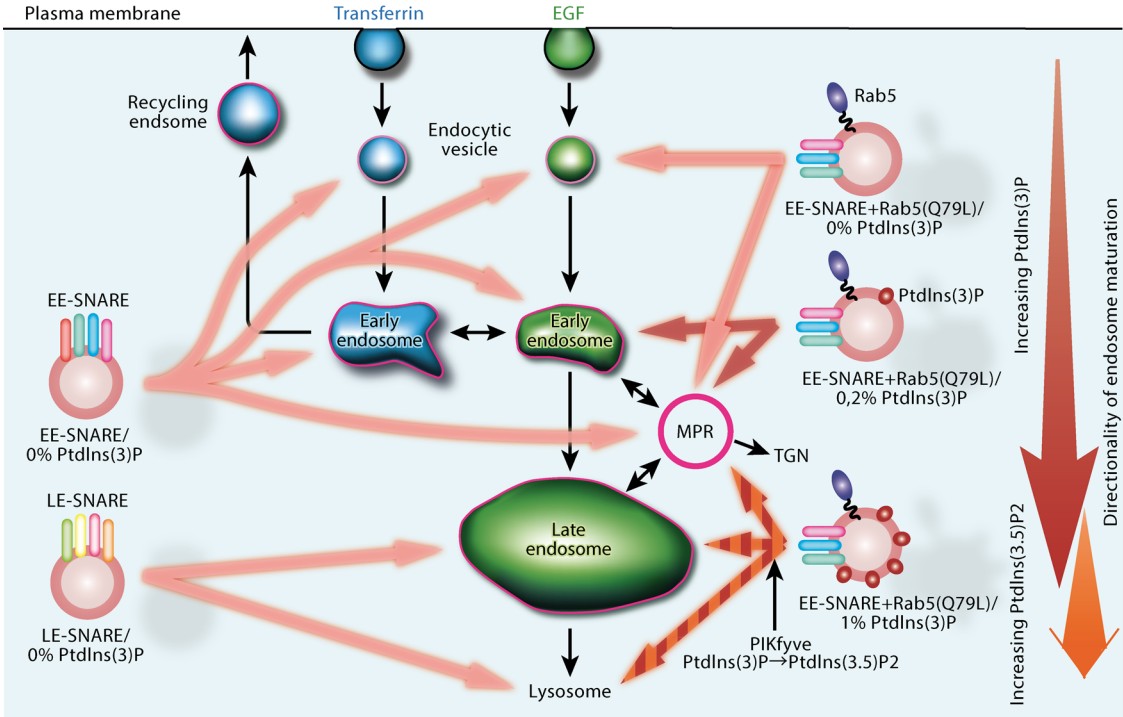

**Fig. 6 | Cartoon summarizing intracellular targeting of injected proteoliposomes.** Early endosomal or late endosomal SNAREs (EE-SNARE or LE-SNARE) are sufficient for a vesicle to be preferentially targeted towards their endogenous counterparts (left side). However, the targets of the liposomes are broader compared to endogenous endosomes. Including various concentrations of phosphoinositides PtdIns(3)P and/or the GTPase Rab5 into the liposome membrane results in the preferential targeting to distinct endosomal subsets (right side). Particularly, the inclusion of GTP-Rab5 on the EE-SNAREs reconstituted liposomes refines targeting only to endocytic vesicles and MPR-positive vesicles (upper right). Upon further inclusion of PtdIns(3)P on the liposomes, targeting specificity is changed by the concentration of PtdIns(3)P. An increase in PtdIns(3)P on the liposomes shifts their targeting from immatured to matured endosomes (see from the top to the bottom of right side). This is consistent with the increasing concentration of PtdIns(3)P upon endosomal maturation. The targeting shift by the concentration of PtdIns(3)P is regulated by the recruitment of PtdIns(3)P binding proteins such as rabenosyn-5 and PIKfyve, with the conversion of PtdIns(3)P into PtdIns(3.5)P2 re-routing the liposomes towards late endosomes. Note that black lines indicate the endogenous endosomal trafficking pathway.

from Jackson ImmunoResearch Laboratories, and HRP-conjugated goat anti-mouse (STAR117P) and goat anti-rabbit (5196–2504) were obtained from Bio-Rad.

## Immunoblotting

HeLa cells were lysed in lysis buffer (50 mM Tris-HCl (pH7.5), 150 mM NaCl, 1% NP-40, 0.5% sodium deoxycholate, 2 mM EDTA, supplemented with protease inhibitor cocktail (Roche)). Lysates were mixed with 2x sample buffer (62.5 mM Tris-HCl (pH6.8), 2% SDS, 10% Glycerol, 5% 2-mercaptethanol, 0.02% Bromophenol blue), electrophoresed on Laemmli SDS-PAGE gel, and transferred onto PVDF membranes (Immobilon-P, Millipore) using a Bio-Rad Trans-Blot system. Membranes were blocked with 1% ECL Prime Blocking Reagent (Cytiva) in TBS (137 mM NaCl, 2.6 mM KCl, 25 mM Tris-HCl (pH7.4)) with 0.1% Triton X-100 for 30 min, followed by incubation with primary antibody in blocking buffer overnight at 4 °C. Horse radish peroxidase-conjugated secondary antibodies were incubated for 60 min at room temperature and developed using ECL prime (Cytiva). Blots were imaged with a LAS 1000 imager (Fujifilm). All Western blots were reproduced at least three times as biological replicates. Source data are provided as a Source Data file.

## Immunostaining

The procedure was described previously[26]. Briefly, HeLa cells were fixed with 4% paraformaldehyde (Sigma-Aldrich) in PBS for 10 min. The cells were then incubated overnight at 4 °C with primary antibody diluted 1:300–1000 in blocking buffer (PBS containing 0.05% saponin and 1% goat serum). After washing in PBS three times, the cells were incubated with the appropriate fluorochrome-conjugated secondary antibodies in blocking buffer for 90 min. Images were acquired using an LSM 780 confocal microscope (Carl Zeiss), using a 63X objective with a numerical aperture of 1.4 with Zen software (Carl Zeiss).

## Preparation of early endosomes and in vitro fusion assay

Early endosomes were prepared from HeLa cells as described[27]. HeLa cells were incubated with 5 μg/ml Alexa Fluor 488- or Alexa Fluor 568-transferrin (Thermo Fisher) for 3 min and were chilled on ice to stop the reaction. These cells were homogenized in homogenization buffer (250 mM sucrose, 3 mM imidazole-HCl, pH 7.4) containing protease inhibitors (Complete EDTA-free, Roche) using a ball homogenizer with a clearance of 0.02 mm[41]. The homogenate was centrifuged at 2000 × g for 15 min, and the postnuclear supernatant (PNS) fraction was layered on top of a Nycodenz gradient consisting of 3 mL each of ice-cold Nycodenz solutions of 28%, 19%, 7.3%, respectively, followed by centrifugation at 200,000 g for 90 min at 4 °C in a Beckman SW41 rotor. The 7.3%/19% boundary (early endosome-rich fraction) was concentrated while changing the buffer to HB150 buffer (150 mM KCl, 20 mM HEPES (pH7.5)) using a VIVASPIN 2 concentrator (30,000 molecular weight cutoff) (Sartorius).

For measuring fusion in vitro, differently labeled endosomes or (proteo)liposomes were mixed with cytosolic fraction from HeLa cells. The samples were incubated at 37 °C by gently shaking for 45 min, the mixtures were added to 24-well plate containing a coverglass coated by poly-L-lysine, followed by centrifugation at 50,000 g for 15 min. The samples were fixed with 4% PFA, and fusion was measured by single

particle counting using a confocal laser microscope. Acquired images were analyzed by Matlab algorithm[40].

To generate cytosol fractions immunodepleted of specific PtdIns(3)P binding proteins, cytosol fractions prepared from HeLa cells were incubated with primary antibody against the respective proteins or control IgG overnight at 4 °C. Protein A Sepharose Fast flow beads (Cytiva) were added to remove the antibody-protein complex from the cytosol.

## Purification of proteins

All SNARE proteins were prepared as described in ref. 29 for early endosome SNAREs (syntaxin 13, vti1a, syntaxin 6, VAMP4), and ref. 30 for late endosome SNAREs (syntaxin 7, vti1b, syntaxin 8, VAMP8). Briefly, the proteins were expressed as His6-tagged or GST-tagged fusion proteins in *Escherichia coli* and purified by Ni2+-agarose (Quiagen) or glutathione-Sepharose (GE Healthcare), respectively. The tags of all proteins were removed by thrombin cleavage. All proteins were further purified by ion-exchange chromatography. Prenylated-Rab5 (WT and Q79L mutant) and prenylated-Rab7 (Q67L mutant) were expressed in Sf9 cells using a baculovirus system and purified by Mono-Q ion exchange column (GE Healthcare)[35]. All proteins were 95% pure, as judged by SDS–PAGE and Coomassie blue staining.

## Preparation of proteoliposomes

Liposome preparation was performed as described[26,27,62]. Basic liposomes consist of 79.7% PC (L-α-phosphatidylcholine), 20% PS (L-α-phosphatidylserine) and, 0.3% Rhodamine-PE (1,2-dioleoyl-*sn*-glycero-3-phosphoethanolamine-*N*-lissamine rhodamine B sulfonyl ammonium salt) (molar ratios). Liposomes containing an endosome lipid composition contain 31% PC, 11% SM (N-stearoyl-D-*erythro*-sphingosylphosphorylcholine), 14% PE (L-α-phosphatidylethanolamine), 3% PS, 40% cholesterol, 1% PtdIns (L-α-phosphatidylinositol) and 0.3% Rhodamine-PE. When PtdIns(3)P was included, the PC concentration was reduced accordingly.

The protein-to-phospholipid molar ratio was adjusted to that of early endosomes[24,63]. Accordingly, for EE-SNARE PL, the protein-to-lipid ratio of syntaxin 13, vti1a, syntaxin 6, Vamp4 was 1:2,000, 1:10,000, 1:1,200, 1:15,400, respectively. For Rab5, we quantified Rab5 and syntaxin 13 in enriched early endosomes prepared from HeLa cells (see above) by quantitative Western blot analysis using purified proteins as a standard, yielding a molar ratio of syntaxin 13: Rab5 (1:0.11), resulting in 1:16,700 as the protein-to-lipid ratio of Rab5. For proteoliposomes containing only two SNAREs, a protein:phospholipid ratio of 1:2,000 was used for each SNARE protein, and for proteoliposomes containing only one SNARE, the ratio was 1:1,000. Finally, 4-LE-SNARE-PL, a ratio of 1:2,000 was used for each of the four SNARE proteins.

## Liposome flotation assay

30% and 80% Nycodenz (Progen) stock solutions were prepared with HP150 buffer. 20 μl 80% (w/w) Nycodenz was added and thoroughly mixed with 20 μl proteoliposomes. Next 40 μl of 30% (w/w) Nycodenz were overlaid and finally 40 μl HP150 buffer was added as top layer. Samples are spun at 275,000 × *g* with the S55S swinging bucket rotor (Hitachi) for 60 min at 4 °C. After centrifugation, 20 μl aliquots are taken from the top of the gradient and analyzed with SDS-PAGE and Western blotting.

## Cell Culture and production of knock-down cell lines

HeLa cells were originally obtained from ATCC (HeLa CCL-2), expanded and cryo-preserved, and then propagated from frozen stocks. They were grown in Dulbecco's modified Eagle medium (DMEM, Lonza GmbH) with the following additions: 10% FCS (Fetal calf serum, PAA laboratories GmbH), 4 mM glutamine (Lenza GmbH), and 100 units/mL each of penicillin and streptomycin (Lonza GmbH). DNA transfection was performed with FuGENE HG transfection reagent (Promega).

Briefly, HeLa cells were grown at a confluence of 60% to 70% in a 24-well plate with a coverglass. Mixture of 0.5 μg of the DNA and 2 μl FuGENE HG reagent in Opti-MEM (Thermo Fisher Scientific) was incubated for 10 min and was then added to the dishes. Sixteen hours post transfection, the medium was changed to DMEM containing 10% FBS.

For the generation of knock-down cell lines for Rab7A, target shRNA containing vectors (GE Dharmacon) were used for transfection, followed by the selection of stable cell lines using 1 μg/ml puromycin (Sigma).

## Microinjection

Microinjection into HeLa cells was performed as described[26,27]. 2 mM lipid (proteo)liposomes, 10 μg/ml DAPI (injection marker) in HB150 were filled in Femtotips (Eppendorf). HeLa cells harvesting 12 mm coverslip was placed into a 35 mm petri dish (Becton Dickinson) filled with pre-warmed culture medium (F12 medium (Invitrogen), supplemented with 10% FCS, 10 mM HEPES (pH7.5) and 100 units/mL each of penicillin and streptomycin). Microinjection was performed using Injectman micromanipulator (Eppendorf) under a Leica DMIL inverted microscope. After microinjection, the cells were incubated for 5 min at 37 °C in the culture medium and then fixed with 4% paraformaldehyde (Sigma-Aldrich) for 10 min followed by immunostaining using antibodies specific for organelles as indicated. To label transferrin-positive endosomes, 5 μg/ml Alexa Fluor 488-Transferrin (final concentrations) were added to the culture medium at the beginning of the injections. After microinjection, the cells were incubated for 5 min at 37 °C and then processed for immunostaining as above. For internalization of Alexa Fluor 488-EGF into endosomes, after HeLa cells were starved in DMEM medium (Lonza) for 3 h, microinjection was performed in 100 ng/ml Alexa Fluor 488-EGF containing injection medium for 5 min and incubated for 5 min at 37 °C in the cell culture medium.

## Image processing

The extent of colocalization between injected proteoliposomes and organelles was determined using a custom written Matlab algorithm (The Mathworks Inc.), kindly provided by Prof. Silvio Rizzoli[41]. At least 100 injected vesicles were analyzed for the colocalization with each organelle marker in an experiment of microinjection.

## Reporting summary

Further information on research design is available in the Nature Portfolio Reporting Summary linked to this article.

# Data availability

The data that support the findings of this study are included in this article, the Supplementary information, with the Source data file containing all data necessary for interpreting, verifying and extending the research. The individual micrographic images used for quantifying colocalizations are available at the repository of the Max-Planck Society (https://edmond.mpg.de/). Source data are provided with this paper.

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

## Acknowledgements
We are grateful to D. Czernik for assistance with mass spectrometry analysis, U. Ries, S. Schmidt, and B. Barg-Kues for technical support, S. Rizzoli for kindly providing the Matlab algorithm for examining the colocalization. We also thank H. Stenmark, and H. Xu, for providing EGFP-2xFYVE, GFP-ML1N*2 cDNA plasmid, respectively. S.K. was supported by the Uehara memorial foundation, Takeda Science Foundation, and JSPS KAKENHI (19K06559).

## Author contributions
S.K. carried out all experiments. S.K. and R.J. designed and discussed the experiments and wrote the manuscript.

## Funding

## Competing interests
The authors declare no competing interests.
