## [Peer Review File · Nature Communications]

Rab GTPases and phosphoinositides fine-tune SNAREs dependent targeting specificity of intracellular vesicle trafficREVIEWER COMMENTS

Reviewer #1 (Remarks to the Author):

This is a beautiful study showing that phosphoinositide lipids (specifically PtdIns(3)P and PtdIns(3,5)P₂) and Rab5 affect the trafficking of vesicles carrying SNARE proteins. Using microinjection of artificial liposomes with reconstituted purified SNARE proteins, the authors show that increasing concentrations of PtdIns(3)P and activation of Rab5 result in more targeting of the liposomes to later endosomal compartments. Moreover, using cell-free fusion assays with purified endosomes and experiments with the PIKfyve inhibitor YM201636, it is shown that this is dependent on the binding of rabenosyn-5 and PIKfyve (which converts PtdIns(3)P to PtdIns(3,5)P₂). These findings for the first time show how the maturation of endosomes along the endocytic pathway can be relatively precisely regulated by only a limited number of factors (one Rab, three SNAREs and two phosphoinositide lipids). I therefore support publication, although I have a number of concerns (mainly in the presentation of the data) that the authors should address:

-My main concern is that only analysed data is shown in the form of bar graphs. In order to assess the quality of the data, the authors should provide representative examples of the underlying microscopy recordings for all conditions, thus microscopy images of the microinjected liposomes, cell-free experiments, immunolabelings and overexpression of tagged proteins.

-My second main comment is that it is too simplistic to call syntaxins 6, 13, Vti1a and VAMP4 “early endosomal” and syntaxins 7, 8, Vti1b, and VAMP8 “late endosomal”. I realize that the Jahn lab has published compelling papers showing that this is the case, but other labs published different (or at least additional) locations for these SNAREs. For example, studies from other labs showed that Vti1a, Vti1b, and VAMP4 mainly locate at the Golgi, whereas VAMP8 has also been reported to locate at recycling endosomes, etc. Whereas this does not affect the quality nor main conclusions of this study, it does affect the interpretation of the results. For example, the finding that the EE-SNARE liposomes locate more to late endosomes does not seem to contradict expectations, but might actually be in line with published observations.

-In figure 2, the authors performed experiments with wildtype Rab5(GDP). However, this form of Rab is not membrane associated, right? It is also missing from the density gradient centrifugation experiments of Supplementary figure 2c. So is this injected as a soluble protein? If so, is a direct comparison with the membrane-bound QL mutant completely warranted? The authors should comment on this.

-Please show the data described in lines 200-207 with fluorescently labelled EGF and Rab11.

-The description of the used constructs is incomplete. What SNAREs were used? The provided citations (26, 27, 60) are not very helpful, as these only refer further to older papers. Reference 60 even seems to concern entirely different SNAREs and does not seem appropriate. Can the authors describe the used SNARE constructs (mouse or human? What isoform? Any tags?) and refer to the original studies where they are described. In addition, the Rab5-GFP, Rab11-GFP, Mon1-GFP and PIKfyve-GFP constructs are not described.

Minor comments:

-In line 131, it is remarked that 20% PS is necessary for efficient reconstitution of proteins. However, in subsequent experiments only 3% PS is used. Please clarify.

-In line 173, it is remarked that only the three Q-SNAREs are reconstituted to avoid liposome clustering. However, this does not seem to affect the experiments of figure 1. Why not?

-In lines 181-184, an explanation is offered for why Rab5 and PtdIns(3)P block EE-SNARE targeting based on VPS13B and GARP/EARP. However, no data on VPS13B and GARP/EARP is shown, so this seems mere speculation. This should be made clear in the text (and perhaps moved to the discussion section)

-I find the title somewhat misleading. To me, the word “overridden” suggests that Rabs and phosphoinositide lipids are the main factors for determining where a vesicle traffics. However, as figure 2B and 2D clearly show, vesicles do not traffic correctly at all without SNAREs. So maybe instead use the word “regulated” or “finetuned”.

-The number of repeats and description of statistical tests are missing for some of the experiments.

Reviewer #2 (Remarks to the Author):

In the manuscript submitted to Nature Communicates entitled “SNAREs dependent targeting specificity of intracellular vesicle traffic can be overridden by Rab GTPases and phosphoinositides” Koike and Jahn rigorously examine how Rabs and phosphoinositides can override targeting specificity of vesicles along

the endosomal system. This is an important and solid study that is well suited for this journal in terms of scope and impact, but it does have some limitations (that can readily be addressed), which are noted below.

Previously in three excellent studies Koike and Jahn lab developed an assay with SNARE specific fluorescently labeled proteoliposome injections to study how the SNARE composition in the proteoliposome can drive the accumulation to specific endogenous compartments (Refs. 7, 26 & 27). Their work identified how the Stx6 and/or Stx13 SNARE can recruit tether factors (i.e. Vps51 and Vps13b) to the injected proteoliposome to regulate the target and fusion to these compartments.

In this manuscript this team leverages their prior advances to explore how the addition of molecular zip-codes Rab5 and/or phosphatidylinositol 3-phosphate (PIP3) affect the targeting of defined microinjected vesicles to different endosomal compartments. The main conclusion of their analysis is: 1) Rab5 and the concentration of PIP3 can refine the targeting of vesicles to different compartment by increasing the recruitment of the Rab5 effectors rabenosyn-5 and PIKfyve and 2) the recruitment of these factors promotes the conversion of PIP3 on the liposome to PI(3,5)P2 that can explain the increase targeting to late endosomal compartments. Both are important findings and refine our understanding how vectoral orchestration of traffic occurs and the hierarchical roles of different motifs. The work supports a complex interplay between Rab5 and PIP3 concentration and conversion at the proteoliposome underscoring how these zip-codes molecules work together to maintain organelle organization and identity.

The fact that they focus in analyzing the targeting of liposome to compartment just 5 minutes after the injection is elegant in that it makes it unlikely that the exogenous proteins and lipid alters the endogenous organelles. Rather the observations and conclusions from their data support cooperative regulation of Rab5 and PIP3.

However, a limitation that they should address is that based on their previous data they should show how the addition of Rab5 or PIP3 may alter the interaction of EE SNARE's liposomes with their interacting partners (e.g. Vps51 or Vps13b) instead of focusing on negative control data. Related, they did an outstanding job demonstrating the specificity of the liposome preparation and validating their results with negative counterpart liposome preparation. However, in my opinion some of the negative control results need not be included as part of the main figures (e.g. Fig 2b and 2d) and rather they should focus in testing other Rab's and other SNARE composition to validate their conclusion regarding Rab5-rabenosyn-5 interaction and PIP3 levels. For example, in Figure 2c when they used EE with Rab5QL, they should also test Rab7QL mutant to see if now it switches from APPL1/MPR to late endosomes? (The Rab5GDP control data in panel Figure 2d could be supplemental.)

To help strengthen their conclusions and validate the importance of Rab5 and PIP3 the authors should consider adding the following additional control experiments (or provide strong justification why they are not needed).

1. On figure 2a they showed that 3-EE-SNARE liposomes with 1% PIP3 switched to late endosomes, but they did not test any other SNAREs. Previously, they showed that liposomes that only contain Stx6 will be highly enriched in the MPR compartment (when compared with 3 or 4-EE-SNARE liposomes). Could the addition of PIP3 to the Stx6 only liposome also switch it to late endosomes? Or are Stx6 SNARE liposomes the only target to MPR? How does the addition of PIP3 alter the interaction of Vps51 or VPs13b to the Stx6 liposomes?
2. On figure 2c they should make a preparation with Rab7QL to see if these liposomes now accumulate in late endosomes compartment instead of MPR or APPL1. This data set should also be analyzed just with Sxt6 liposomes with Rab5QL and Rab7QL.
3. Figure 3 support that the addition of Rab5QL and 1% PIP3 now is highly enriched in LBPA and LAMP1, but not Rab5-GDP. Here a Rab7QL experiment would be a better experiment than Rab5-GDP. Also analyzing a liposome with just Stx6 to show that the addition of Rab5QL and PIP3 are the main reason for the switching, and this depend on the recruitment of the proper Rab5 effectors.
4. Figure 4d in-vitro fusion assays clearly demonstrated that the cytosol immunodepleted Rabenosyn-5 is part of the inhibition observed. However, they did not show that 3EE-SNARE's Rab5QL 1% PIP3 can interact with Rabenosyn-5 - as they showed for PIKfyve or in previous papers on Vps51 and VPs13b. They should perform the same pulldown experiment they did for PIKfyve on Figure 5b with Rabenosyn-5 to demonstrate that it also increases with more PIP3 on the liposomes. In addition, in Supplementary Figure S2d they showed endogenous Rab5 colocalizing with the liposomes. The same experiment should be done for Rabenosyn-5 to demonstrate that the endogenous Rabenosyn-5 colocalizes with the liposomes only when the liposomes contain Rab5QL and 1% PIP3.
5. The pulldown experiments in Figure 5b support that PIKfyve interaction with the liposome depends on PIP3 levels, and this will explain the switch to PI(3,5)P2. However, previously they showed that Vps13b also bind PIP3 (albeit with 5% PIP3 versus 1% here). As the PIKfyve interaction increases with SNARE-Rab5QL-PIP3 liposome, what happen with Vps13b or Vps51 interaction under this condition? If these Vps proteins can no longer be recruited to the liposomes this may explain the late-endosomal targeting.

Minor comments:

1. Figure 1. Fig. 1 is really a control experiment for their previous published data and is not necessary as part of the main figures (rather could be supplemental).
2. Figure 6. The cartoon in Fig. 6 is rather busy and confusing, especially as there are 3 different set of arrows (large ones, black lines, orange ones) making it hard to follow. The authors should consider showing the orange arrows differently. The right side of the model is from their previously publish data and may not be necessary.

3. Labels

- a. The vesicle diagram on the figure is showing 4 SNAREs but the legend and results said they used the 3 SNARE's combination.
- b. The y-axis scale on panel c should be the same to panel a (0-6).
- c. Some references format on the text are not superscripted (line 158, 175, 188).

Reviewer #3 (Remarks to the Author):

In this article, the authors used a robust assay, which they previously pioneered, to inject cells with liposomes of controlled lipid and protein content and follow their fate towards intracellular membrane compartments. Here they tested the role of early vs late endosomal v-/t-SNAREs, PI3P vs PIP2, and the absence or presence of Rab5, in its GDP inactive or GTP active form. They completed their study using in vitro assays with liposomes and endosomes extracted from cells.

The experiments were very rigorously carried out and controlled. The results are novel and of high interest to understanding the specificity of intracellular membrane fusion, particularly how SNAREs, Rabs and PIs could collaborate in this process.

The main finding is that PI3P loaded at high concentration onto liposomes also carrying early endosomal SNAREs and active rab5 is converted into PIP2 by PIKfyve leading to routing to late endosomes. This was prevented by pharmacological inhibition of the conversion of PI3P. At low concentrations or in the absence of PI3P and in the absence of Rab5, however, the authors observed a routing of the liposomes to early endosomes, particularly those containing M6PR.

While this insightful experimental approach proves here to be powerful, it also opens a large array of questions, some of which will require very substantial additional work in future studies to understand the role of other Rabs (4, 11 in particular), SNAREs and PIPx species, this reviewer still thinks that some clarifications are needed, which might require additional experiments.

In particular, the authors used in most experiments a full set of v- and t-SNAREs in their liposomes, whereas in others they only had t-SNAREs (or Q-SNAREs as expressed by the authors). The rationale is not clear and a direct comparison of both combinations in the most critical experiments would help to understand the difference. Along the same line, when VAMP4 or VAMP8 is omitted from the liposomes, the authors could test to which endogenous VAMPx-containing compartment the liposomes would be

routed. That might help understand the routing to MPR endosomes (aren't they rich in VAMP4). Also, what would happen to liposomes only carrying VAMP4 or VAMP8, PI3P and Rab5?

Rather than an overridden SNARE-dependent targeting specificity, this reviewer would like to suggest to rather write in the title that the findings of the authors suggest a restriction of that specificity.

Minor comments:

Along with the suggestion abovementioned, it would help to add the precise SNARE composition of liposomes used in each figure, in the corresponding legend.

Reviewer #4 (Remarks to the Author):

The manuscript of Koike and Jahn focuses on the SNARE-dependent targeting of liposomes to endosomes. For this, the authors used two different approaches. They generated liposomes with endosomal SNAREs and decorated them with selected phosphoinositides and Rab5. They then injected these into HeLa cells and monitored colocalization with selected early endosomal and late endosomal marker. Furthermore, the authors used these vesicles to monitor in vitro fusion with endosomes using cytosol, which was either untreated or depleted of selected proteins. They observe that PI3P binding and PI3P to PI(3,5)P2 conversion were sufficient to override the Rab5 specificity and early endosomal SNAREs, suggesting a hierarchy of endosomal proteins in determining targeting to early and late endosomes.

This is an overall quite convincing manuscript with many controls, which highlight the importance of PIPs in specificity of targeting of vesicles. I found in particular the observation that high PI3P redirected liposomes very intriguing, as this suggests that the PIP content overrides the remaining targeting information on liposomes. However, there are a few issues that require attention or clarification:

1. The colocalization index is not clear to me. Is normalized colocalization reflecting that just 1-5% of the injected liposomes get targeted to the respective compartment? Please clarify.
2. The authors use liposomes with Rab5 and SNAREs and PIPs, but conclude that the PIP content overrides the targeting. In Figure 5b, c and d, the authors now use liposomes that always carry SNAREs in addition to PIPs. What happens, if they do the experiment with liposomes lacking SNAREs on their surface? Is the PIP information together with Rab5-GTP sufficient for targeting?

3. The authors may want to read up on the Mon1-Ccz1 literature regarding Rab7 activation. Its function as a GEF was solved by Nordmann et al., 2010 and Gerondopoulos et al., 2012, and the Rab5-dependent targeting and activation was not resolved by the Poteryaev study, but by Langemeyer et al., 2020.

Koike and Jahn

Detailed response to the comments of the reviewers

Reviewer #1 (Remarks to the Author):

This is a beautiful study showing that phosphoinositide lipids (specifically PtdIns(3)P and PtdIns(3,5)P₂) and Rab5 affect the trafficking of vesicles carrying SNARE proteins. Using microinjection of artificial liposomes with reconstituted purified SNARE proteins, the authors show that increasing concentrations of PtdIns(3)P and activation of Rab5 result in more targeting of the liposomes to later endosomal compartments. Moreover, using cell-free fusion assays with purified endosomes and experiments with the PIKfyve inhibitor YM201636, it is shown that this is dependent on the binding of rabenosyn-5 and PIKfyve (which converts PtdIns(3)P to PtdIns(3,5)P₂). These findings for the first time show how the maturation of endosomes along the endocytic pathway can be relatively precisely regulated by only a limited number of factors (one Rab, three SNAREs and two phosphoinositide lipids). I therefore support publication, although I have a number of concerns (mainly in the presentation of the data) that the authors should address:

My main concern is that only analysed data is shown in the form of bar graphs. In order to assess the quality of the data, the authors should provide representative examples of the underlying microscopy recordings for all conditions, thus microscopy images of the microinjected liposomes, cell-free experiments, immunolabelings and overexpression of tagged proteins.

As requested by the referee, we have now included exemplary images of several experiments into the Supplemental Material (see Fig. S1c, Fig. S2c, Fig. S3e, Fig. S4b, Fig. S6a, and Fig. S6b). Please note that this is our third publication using microinjection of labeled liposomes into living cells, with a more detailed description of the procedure and sets of control experiments being published previously (Koike and Jahn, 2017, Koike and Jahn, 2019, as cited). The procedure (including image processing and the analysis of co-localization) is described in detail in these papers and is highly standardized, reproducible and reliable.

My second main comment is that it is too simplistic to call syntaxins 6, 13, Vti1a and VAMP4 “early endosomal” and syntaxins 7, 8, Vti1b, and VAMP8 “late endosomal”. I realize that the Jahn lab has published compelling papers showing that this is the case, but other labs published different (or at least additional) locations for these SNAREs. For example, studies from other labs showed that Vti1a, Vti1b, and VAMP4 mainly locate at the Golgi, whereas VAMP8 has also been reported to locate at recycling endosomes, etc. Whereas this does not affect the quality nor main conclusions of this study, it does affect the interpretation of the results. For example, the finding that the EE-SNARE liposomes locate more to late endosomes does not seem to contradict expectations, but might actually be in line with published observations.

We are obviously aware of the fact that sets of SNAREs may operate in more than one intracellular fusion step, and furthermore, that for a given individual fusion step is not always clear to which extent one particular SNARE may be replaced by another of the same subfamily (with such “promiscuity” apparently more common in the endocytic limb of the secretory pathway than in other trafficking steps). Also, as shown previously by us and others, SNARE proteins generally exhibit a broader distribution compared to other organelle identity molecules such as phosphoinositides and Rab GTPases because most are membrane-anchored for their entire lifespan and thus need to be recycled by vesicular trafficking after completion of a fusion reaction. Thus, the localization of a SNARE protein to a given organelle is not sufficient to conclude that the SNARE functions in a fusion step of this organelle – it may merely be a passenger cycling back to its site of action. While we acknowledge that there is still some discussion which SNAREs are indeed mediating fusion of early and late endosomes in cells (e.g. syntaxin 13 vs. syntaxin 16), the evidence for the complex discussed here to be mainly involved in early endosome fusion is quite strong overall. We have modified the introduction (page 4, line 3) to acknowledge these issues but we ask for your understanding that we have retained the nomenclature for convenience.

In figure 2, the authors performed experiments with wildtype Rab5(GDP). However, this form of Rab is not membrane associated, right? It is also missing from the density gradient centrifugation experiments of Supplementary figure 2c. So is this injected as a soluble protein? If so, is a direct comparison with the membrane-bound QL mutant completely warranted? The authors should comment on this.

We have used prenylated Rab5 for all experiments, and accordingly GDP-bound Rab5 was inserted in the liposome membrane during reconstitution. To show this, we performed a flotation assay to separate the liposomes from non-incorporated proteins, now included in the supplemental material (Fig. S3b). Obviously, after injection it is likely that GDP-Rab5 is removed from the membrane by cytoplasmic GDI.

Please show the data described in lines 200-207 with fluorescently labelled EGF and Rab11.

We apologize for not repeating the reference to Fig. 3b in the text describing these experiments which may have given rise to a misunderstanding – these data are (and were in the original version) included in the figure, and we have now added the reference to the figure.

The description of the used constructs is incomplete. What SNAREs were used? The provided citations (26, 27, 60) are not very helpful, as these only refer further to older papers. Reference 60 even seems to concern entirely different SNAREs and does not seem appropriate. Can the authors describe the used SNARE constructs (mouse or human? What isoform? Any tags?) and refer to the original studies where they are described. In addition, the Rab5-GFP, Rab11-GFP, Mon1-GFP and PIKfyve-GFP constructs are not described.

We have added a new section in “Materials and Experimental Procedures” to describe the DNA constructs used in this study in more detail, and we have checked the references (page 13, line 464).

Minor comments

- In line 131, it is remarked that 20% PS is necessary for efficient reconstitution of proteins. However, in subsequent experiments only 3% PS is used. Please clarify.

We apologize for this misleading remark that has been corrected. SNARE proteins cannot be efficiently reconstituted in liposomes containing 100 % PC (phosphatidylcholine) but it works when 20% PS is included. On the other hand, only 3% PS are sufficient for reconstitution when an endosomal lipid composition is used (see Fig. S2b).

-In line 173, it is remarked that only the three Q-SNAREs are reconstituted to avoid liposome clustering. However, this does not seem to affect the experiments of figure 1. Why not?

Thank you for pointing out this issue that requires additional information to be understandable. When we co-reconstituted 4-SNARE proteins together with the Rab proteins using 1% PtdIns(3)P liposomes, they tended to aggregate, which is not observed in the absence of Rabs. This aggregation resulted in the clogging of the microinjection needle. To solve this issue, we omitted the R-SNARE Vamp4 from the reconstitution. As shown in our previous paper, liposomes containing the three Q-SNAREs (Syntaxin 6, Syntaxin 13, Vti1a) showed very similar targeting to liposomes containing all four EE SNAREs (Syntaxin 6, Syntaxin 13, Vti1a, Vamp4). Moreover, liposomes containing only Vamp4 showed no specific targeting suggesting that it is not involved in targeting (Koike and Jahn, 2019, Nature communications). We have added an explanatory sentence during revision (Page 6, line 202).

In lines 181-184, an explanation is offered for why Rab5 and PtdIns(3)P block EE-SNARE targeting based on VPS13B and GARP/EARP. However, no data on VPS13B and GARP/EARP is shown, so this seems mere speculation. This should be made clear in the text (and perhaps moved to the discussion section)

We agree and have removed the sentences.

I find the title somewhat misleading. To me, the word “overridden” suggests that Rabs and phosphoinositide lipids are the main factors for determining where a vesicle traffics. However, as figure 2B and 2D clearly show, vesicles do not traffic correctly at all without SNAREs. So maybe instead use the word “regulated” or “finetuned”.

Thank you for the helpful suggestion, which was also brought up by Referee 3, and we have modified the title accordingly.

The number of repeats and description of statistical tests are missing for some of the experiments.

We have stated the number of experimental samples and independent repeats in all the figure legends. Since the number of repetitions was variable, we have indicated each datapoint derived from an independent measurement by a black dot in the bar diagrams. The statistical tests are now explained in the legends.

Reviewer #2 (Remarks to the Author):

In the manuscript submitted to Nature Communicates entitled “SNAREs dependent targeting specificity of intracellular vesicle traffic can be overridden by Rab GTPases and phosphoinositides” Koike and Jahn rigorously examine how Rabs and phosphoinositides can override targeting specificity of vesicles along the endosomal system. This is an important and solid study that is well suited for this journal in terms of scope and impact, but it does have some limitations (that can readily be addressed), which are noted below.

We thank the reviewer for the overall positive assessment of our work

Previously in three excellent studies Koike and Jahn lab developed an assay with SNARE specific fluorescently labeled proteoliposome injections to study how the SNARE composition in the proteoliposome can drive the accumulation to specific endogenous compartments (Refs. 7, 26 & 27). Their work identified how the Stx6 and/or Stx13 SNARE can recruit tether factors (i.e. Vps51 and Vps13b) to the injected proteoliposome to regulate the target and fusion to these compartments.

In this manuscript this team leverages their prior advances to explore how the addition of molecular zip-codes Rab5 and/or phosphatidylinositol 3-phosphate (PIP3) affect the targeting of defined microinjected vesicles to different endosomal compartments. The main conclusion of their analysis is: 1) Rab5 and the concentration of PIP3 can refine the targeting of vesicles to different compartment by increasing the recruitment of the Rab5 effectors rabenosyn-5 and PIKfyve and 2) the recruitment of these factors promotes the conversion of PIP3 on the liposome to PI(3,5)P2 that can explain the increase targeting to late endosomal compartments. Both are important findings and refine our understanding how vectoral orchestration of traffic occurs and the hierarchical roles of different motifs. The work supports a complex interplay between Rab5 and PIP3 concentration and conversion at the proteoliposome underscoring how these zip-codes molecules work together to maintain organelle organization and identity.

The fact that they focus in analyzing the targeting of liposome to compartment just 5 minutes after the injection is elegant in that it makes it unlikely that the exogenous proteins and lipid alters the endogenous organelles. Rather the observations and conclusions from their data support cooperative regulation of Rab5 and PIP3.

Again, we appreciate the positive comments of the reviewer

However, a limitation that they should address is that based on their previous data they should show how the addition of Rab5 or PIP3 may alter the interaction of EE SNARE's liposomes with their interacting partners (e.g. Vps51 or Vps13b) instead of focusing on negative control data.

We agree that these are very interesting questions, and some of them have been addressed in new microinjection experiments that are described in more detail below.

Related, they did an outstanding job demonstrating the specificity of the liposome preparation and validating their results with negative counterpart liposome preparation. However, in my opinion some of the negative control results need not be included as part of the main figures (e.g. Fig 2b and 2d)

As requested, these panels have now been moved to Supplemental (Fig. S2d, Fig. S3g, Fig. S5h)

and rather they should focus in testing other Rab's and other SNARE composition to validate their conclusion regarding Rab5-rabenosyn-5 interaction and PIP3 levels. For example, in Figure 2c when they used EE with Rab5QL, they should also test Rab7QL mutant to see if now it switches from APPL1/MPR to late endosomes? (The Rab5GDP control data in panel Figure 2d could be supplemental.)

see below – these experiments have now been included.

To help strengthen their conclusions and validate the importance of Rab5 and PIP3 the authors should consider adding the following additional control experiments (or provide strong justification why they are not needed).

1. On figure 2a they showed that 3-EE-SNARE liposomes with 1% PIP3 switched to late endosomes, but they did not test any other SNAREs. Previously, they showed that liposomes that only contain Stx6 will be highly enriched in the MPR compartment (when compared with 3 or 4-EE-SNARE liposomes). Could the addition of PIP3 to the Stx6 only liposome also switch it to late (when compared with 3 or 4-EE-SNARE liposomes) late endosomes? Or are Stx6 SNARE liposomes the only target to MPR? How does the addition of PIP3 alter the interaction of Vps51 or VPs13b to the Stx6 liposomes?

As suggested by the reviewer, we conducted new microinjection experiments using syntaxin 6 liposomes containing 1% PtdIns(3)P and compared their targeting with those reconstituted with 4-EE-SNAREs and 1% PtdIns(3)P (new Figure 2b). Interestingly, we observed that both sets of liposomes were targeted to MPR-positive compartments and lysosomes, i.e. the inclusion of 1% PtdIns(3)P did not cause any major change in the targeting specificity of syntaxin 6 liposomes (except, perhaps, for an additional

preference for lysosomes). Intriguingly, recruitment of *Vps51* to syntaxin 6 liposomes is not affected by the presence of *PtdIns(3)P* (new Fig. S2e), suggesting that *Vps51* and the factors mediating *PtdIns(3)P*-dependent targeting to lysosomes operate in an additive fashion.

2. On figure 2c they should make a preparation with Rab7QL to see if these liposomes now accumulate in late endosomes compartment instead of MPR or APPL1. This data set should also be analyzed just with Sxt6 liposomes with Rab5QL and Rab7QL.

As requested by the reviewer, we conducted new microinjection experiments using 4-EE-SNAREs and Rab7(Q67L) liposomes and added the results to Figure 2c. These experiments show that the inhibitory effects of Rab5(Q79L) on SNARE-dependent targeting for early endosomes were specific, as Rab7(Q67L) did not show the same effects. However, as one may have predicted, colocalization with LAMP1 was increased.

3. Figure 3 support that the addition of Rab5QL and 1% PIP3 now is highly enriched in LBPA and LAMP1, but not Rab5-GDP. Here a Rab7QL experiment would be a better experiment than Rab5-GDP. Also analyzing a liposome with just Stx6 to show that the addition of Rab5QL and PIP3 are the main reason for the switching, and this depend on the recruitment of the proper Rab5 effectors.

*As suggested, we conducted new microinjection experiments using 3-EE-SNAREs and Rab7(Q67L) liposomes containing 0.2% or 1% *PtdIns(3)P*, with the new data shown in new Fig. 3c. At 0.2% *PtdIns(3)P* concentrations, 3-EE-SNAREs and Rab5(Q79L) liposomes showed high colocalization with EEA1 (Fig. 3c) but this was lost when Rab5(Q79L) was replaced with Rab7(Q67L). At 1% *PtdIns(3)P* concentration, liposomes containing 3-EE-SNAREs and either Rab5(Q79L) or Rab7(Q67L) were colocalized mainly with LBPA and LAMP1 (Fig. 3c). The latter is to be expected considering that Rab7 promotes fusion of late endosomes.*

*Following the suggestion of the referee, we also reduced the number of reconstituted SNARE proteins using liposomes containing 0.2% *PtdIns(3)P* and Rab5(Q79L) and analyzed their targeting after microinjection. Interestingly, targeting to the EEA1 compartment was preserved in the presence of both syntaxin 6 and syntaxin 13 but lost when only syntaxin 6 was present (new Fig. 3d). When the *PtdIns(3)P* concentration was increased to 1%, syntaxin 6 and Rab5(Q79L) liposomes were targeted to LAMP1-positive lysosomes, similar to 3-EE-SNARE liposomes (Fig. 3b, and not shown), suggesting that syntaxin 6 alone is sufficient for the lysosomal targeting of liposomes reconstituted with SNAREs, Rab5(Q79L) and 1% *PtdIns(3)P*.*

4. Figure 4d in-vitro fusion assays clearly demonstrated that the cytosol immunodepleted Rabenosyn-5 is part of the inhibition observed. However, they did not show that 3EE-SNARE's Rab5QL 1% PIP3 can interact with Rabenosyn-5 - as they showed for PIKfyve or in previous papers on *Vps51* and *VPs13b*. They should perform the same pulldown experiment they did for PIKfyve on Figure 5b with Rabenosyn-5 to

demonstrate that it also increases with more PIP3 on the liposomes. In addition, in Supplementary Figure S2d they showed endogenous Rab5 colocalizing with the liposomes. The same experiment should be done for Rabenosyn-5 to demonstrate that the endogenous Rabenosyn-5 colocalizes with the liposomes only when the liposomes contain Rab5QL and 1% PIP3.

To address this issue, we performed a microscopy-based assay to measure the recruitment of GFP-Rabenosyn-5 on the liposomes, which increased with the PtdIns(3)P concentration in vitro (Fig. 4h). Interestingly, the recruitment was saturated already at 0.2% PtdIns(3)P, suggesting that the opposite effects of Rabenosyn-5 on the targeting of SNARE liposomes to EEA1-positive endosomes, depending on PtdIns(3)P concentration, do not result from differences in the Rabenosyn-5 concentration on the membrane. In addition, we quantified the fluorescence intensity of GFP-Rabenosyn-5 on the injected liposomes and endogenous endosomes, which were found to be comparable, suggesting that Rabenosyn-5 was recruited on the liposomes and endosomes in cells at the same level (Fig. 4f and 4g).

5. The pulldown experiments in Figure 5b support that PIKfyve interaction with the liposome depends on PIP3 levels, and this will explain the switch to PI(3,5)P2. However, previously they showed that Vps13b also bind PIP3 (albeit with 5% PIP3 versus 1% here). As the PIKfyve interaction increases with SNARE-Rab5QL-PIP3 liposome, what happen with Vps13b or Vps51 interaction under this condition? If these Vps proteins can no longer be recruited to the liposomes this may explain the late-endosomal targeting.

To address this point at least to some extent, we examined the effects of PtdIns(3)P concentration on the Vps51 interaction (Fig. S2e). Our data show that Vps51 interaction is independent of PtdIns(3)P on the membrane.

Minor comments:

1. Figure 1. is really a control experiment for their previous published data and is not necessary as part of the main figures (rather could be supplemental).

Previous Fig. 1a showed a cartoon depicting the organelle markers used in this study, and a similar cartoon was already shown in our previous paper (Koike and Jahn, 2019). Therefore, we moved this panel to Supplemental Figure 1. However, we disagree that the experiment shown in Fig. 1b (now Fig. 1a) is merely a control experiment. In this manuscript we show that inclusion of Rabs or PIPs modify the targeting specificity by SNAREs, particularly by selective inhibition. In contrast, inclusion of a higher SNARE diversity does not result in selective inhibition but rather appears to be additive, i.e. they do not restrict targeting specificity as seen with the Rabs and PtdIns-phosphates. Thus, we would like to retain the figure.

2. Figure 6. The cartoon in Fig. 6 is rather busy and confusing, especially as there are 3 different set of arrows (large ones, black lines, orange ones) making it hard to follow.

The authors should consider showing the orange arrows differently. The right side of the model is from their previously published data and may not be necessary.

We beg to disagree. The black arrows connect the steps of the canonical endocytic pathway, leading from the plasma membrane via endosomal intermediates to the lysosome. On the left and right we show targeting specificity of injected liposomes that contain either only SNAREs (left, published previously) and SNAREs containing GTP-Rab5 and increasing concentrations of PtdIns(3)P. The main conclusions are indeed derived from the comparison between the different sets of liposomes (including those we published some years ago), and therefore we have simplified it a bit, but would like to retain the figure as it shows the main results in one diagram as a “take-home” message.

3. Labels

a. The vesicle diagram on the figure is showing 4 SNAREs but the legend and results said they used the 3 SNARE's combination.

We apologize for our mistake. Our explanation in the legend contained an error. We reconstituted 4-EE-SNAREs on liposomes containing 1% PtdIns(3)P. We have corrected the mistake.

b. The y-axis scale on panel c should be the same to panel a (0-6).

To facilitate comparison between panels, we have used the same scale on the y-axis on all panels in Fig.2.

c. Some references format on the text are not superscripted (line 158, 175,188).

Thank you for your suggestion. We have corrected the format.

Reviewer #3 (Remarks to the Author):

In this article, the authors used a robust assay, which they previously pioneered, to inject cells with liposomes of controlled lipid and protein content and follow their fate towards intracellular membrane compartments. Here they tested the role of early vs late endosomal v-/t-SNAREs, PI3P vs PIP2, and the absence or presence of Rab5, in its GDP inactive or GTP active form. They completed their study using in vitro assays with liposomes and endosomes extracted from cells.

The experiments were very rigorously carried out and controlled. The results are novel and of high interest to understanding the specificity of intracellular membrane fusion, particularly how SNAREs, Rabs and PIs could collaborate in this process.

We appreciate the overall positive assessment of our work

The main finding is that PI3P loaded at high concentration onto liposomes also carrying early endosomal SNAREs and active rab5 is converted into PIP2 by PIKfyve leading to routing to late endosomes. This was prevented by pharmacological inhibition of the conversion of PI3P. At low concentrations or in the absence of PI3P and in the absence of Rab5, however, the authors observed a routing of the liposomes to early endosomes, particularly those containing M6PR.

While this insightful experimental approach proves here to be powerful, it also opens a large array of questions, some of which will require very substantial additional work in future studies to understand the role of other Rabs (4, 11 in particular), SNAREs and PIPx species, this reviewer still thinks that some clarifications are needed, which might require additional experiments.

In particular, the authors used in most experiments a full set of v- and t-SNAREs in their liposomes, whereas in others they only had t-SNAREs (or Q-SNAREs as expressed by the authors). The rationale is not clear and a direct comparison of both combinations in the most critical experiments would help to understand the difference. Along the same line, when VAMP4 or VAMP8 is omitted from the liposomes, the authors could test to which endogenous VAMPx-containing compartment the liposomes would be routed. That might help understand the routing to MPR endosomes (aren't they rich in VAMP4). Also, what would happen to liposomes only carrying VAMP4 or VAMP8, PI3P and Rab5?

We apologize for the lack of clarity. In our previous studies we characterized the role of individual SNAREs in detail, which showed, for instance, that Vamp4 does not have any influence on targeting and does not possess any targeting signal on its own (Koike and Jahn, 2019, Nature communications), which is also true in the presence of PtdInsPx (unpublished observation). We have included an explanation in the text for better clarity (Page 6 line 202). Vamp4 was omitted in the co-reconstitution experiments with Rab5 due to technical problems, which are explained in our response to Reviewer 1 above.

Note that it is still unclear how the specificity of SNARE pairing is guaranteed in cells when considering that no such specificity is observable in vitro. The data presented in this paper show that specificity can be overridden by Rab GTPase and phosphoinositides, suggesting that pairing specificity between Q-SNAREs and R-SNARE is not sufficient to explain the specificity of targeting and fusion of trafficking vesicles (see Koike and Jahn (2022) for a more detailed discussion).

Rather than an overridden SNARE-dependent targeting specificity, this reviewer would like to suggest to rather write in the title that the findings of the authors suggest a restriction of that specificity.

This was also suggested by Reviewer 1, and we have therefore changed the title of the paper accordingly.

Minor comments:

Along with the suggestion above mentioned, it would help to add the precise SNARE composition of liposomes used in each figure, in the corresponding legend.

We appreciate the valuable suggestion. As requested, we have added cartoons of liposomes and the names of the reconstituted SNAREs to all the graphs for easier understanding.

Reviewer #4 (Remarks to the Author):

The manuscript of Koike and Jahn focuses on the SNARE-dependent targeting of liposomes to endosomes. For this, the authors used two different approaches. They generated liposomes with endosomal SNAREs and decorated them with selected phosphoinositides and Rab5. They then injected these into HeLa cells and monitored colocalization with selected early endosomal and late endosomal marker. Furthermore, the authors used these vesicles to monitor in vitro fusion with endosomes using cytosol, which was either untreated or depleted of selected proteins. They observe that PI3P binding and PI3P to PI(3,5)P2 conversion were sufficient to override the Rab5 specificity and early endosomal SNAREs, suggesting a hierarchy of endosomal proteins in determining targeting to early and late endosomes.

This is an overall quite convincing manuscript with many controls, which highlight the importance of PIPs in specificity of targeting of vesicles. I found in particular the observation that high PI3P redirected liposomes very intriguing, as this suggests that the PIP content overrides the remaining targeting information on liposomes.

We thank the reviewer for the positive assessment of our work

However, there are a few issues that require attention or clarification:

1. The colocalization index is not clear to me. Is normalized colocalization reflecting that just 1-5% of the injected liposomes get targeted to the respective compartment? Please clarify.

The reviewer may be concerned about the low colocalization rate. However, as we previously reported (Koike and Jahn, 2017), liposomes were captured by the Golgi with variable delay times after injection. In contrast, SNARE dependent targeting was observed already 5 min after injection. Therefore, in this and our previous papers, to quantify the targeting effects by SNARE proteins, we examined liposomal targeting just 5 min after microinjection. Therefore, the degree of colocalization appears to be relatively low. In our previous study, we also injected endosomes isolated by subcellular fractionation as positive control, reaching co-localizations of 11-16%, which serves as reference for our measurements.

2. The authors use liposomes with Rab5 and SNAREs and PIPs, but conclude that the PIP content overrides the targeting. In Figure 5b, c and d, the authors now use

liposomes that always carry SNAREs in addition to PIPs. What happens, if they do the experiment with liposomes lacking SNAREs on their surface? Is the PIP information together with Rab5-GTP sufficient for targeting?

The targeting of PtdIns(3)P- or Rab5-containing liposomes is shown in Fig. S2d and Fig. S3g, respectively. In summary, in the absence of SNAREs these liposomes were not significantly colocalized with any specific organelles. Note that we define colocalization by the minimum distance between injected liposomes and closest organelles, which needs to be within 100 nm (see Koike and Jahn, 2017, for details), which, as shown previously, indicates fusion. However, it is possible that liposomes containing Rabs and PtdInsPx dock to their targets but do not fuse if SNAREs are lacking. Indeed, the histogram of the minimum distance between injected 1% PtdIns(3)P containing liposomes to EEA1-positive endosomes has a broad peak around 300 nm compared to no PtdIns(3)P containing liposomes (see Figure A). Quantification showed that PtdIns(3)P (0.05-1%) containing liposomes significantly accumulated adjacent to EEA1-positive endosomes but not to other compartments (Figure B), suggesting that PtdIns(3)P induces some degree of selective docking with the EEA1-compartment. To substantiate that the minimum distance criterium used here to measure docking is indeed valid, additional experiments are required that go beyond the scope of our work. We have therefore not included these observations in this paper.

3. The authors may want to read up on the Mon1-Ccz1 literature regarding Rab7 activation. Its function as a GEF was solved by Nordmann et al., 2010 and Gerondopoulos et al., 2012, and the Rab5-dependent targeting and activation was not resolved by the Poteryaev study, but by Langemeyer et al., 2020.

We apologize for this oversight and have now included the relevant references and, in the light of the new findings, discussed possible reasons for the lack of Mon1-Ccz1

complex binding in our system in Discussion (page 11, line 405).

REVIEWERS' COMMENTS

Reviewer #1 (Remarks to the Author):

The authors have addressed all my comments. This is a really interesting and important study, and I endorse publication.

Reviewer #2 (Remarks to the Author):

In the revision the authors addressed through experiments nearly all of my critiques. Thank you.

I have only two very minor suggestions on the revision to help improve the manuscript:

Major POINT #4: The authors add four panels in 4f to show the localization, but they draw an arrow to show the line profile. the line is on top of where the liposome and Rabenosyn colocalize and the panels are too small to see the said colocalization. They could make an inset or highlight other liposome with colocalization.

MINOR POINT #2: They should mention the meaning of the black arrow as well as the rest of the info - either in the legend or the main text. On the discussion line 350 they referenced to this model as "A cartoon summarizing our findings" They never refer to the cartoon again and the current legend is too simplistic.

Otherwise this really is a TERRIFIC paper.

Reviewer #3 (Remarks to the Author):

The authors have VERY satisfactorily answered the reviewers' comments.

Reviewer #4 (Remarks to the Author):

The authors addressed all my questions. I thus have no further experimental requests.

Just one comment regarding Mon1-Ccz1 and Rab7. The authors do not find GFP-Mon1 on their liposomes, regardless of their concentration of PI3P. I would not draw too many conclusions on this. Mon1-Ccz1 functions as a dimer and there is no evidence that the monomer, which is prevalent upon over expression, will do anything. The interaction with Rabex5 and HOPS are purely based on yeast-two-hybrid interactions. While there could be a function in recruitment, I find this rather unlikely. It is just presently unclear. In vitro data clearly show how the complex gets to membranes, and the mutagenesis supports the findings of Kümmel and Ungermann.

The authors can leave this in the discussion as it is, but I would suggest to put a qualifier into their statement as they did not really look at the endogenous complex, and do not even know if the GFP-Mon1 is functional in their assay.